# Intelligent wearable allows out-of-the-lab tracking of developing motor abilities in infants

Manu Airaksinen [1✉], Anastasia Gallen [1], Anna Kivi[1,2], Pavithra Vijayakrishnan [1], Taru Häyrinen[1,2], Elina Ilén[3], Okko Räsänen[4], Leena M. Haataja[1,2] & Sampsa Vanhatalo [1,5✉]

## Abstract

**Background** Early neurodevelopmental care needs better, effective and objective solutions for assessing infants' motor abilities. Novel wearable technology opens possibilities for characterizing spontaneous movement behavior. This work seeks to construct and validate a generalizable, scalable, and effective method to measure infants' spontaneous motor abilities across all motor milestones from lying supine to fluent walking.

**Methods** A multi-sensor infant wearable was constructed, and 59 infants (age 5–19 months) were recorded during their spontaneous play. A novel gross motor description scheme was used for human visual classification of postures and movements at a second-level time resolution. A deep learning -based classifier was then trained to mimic human annotations, and aggregated recording-level outputs were used to provide posture- and movement-specific developmental trajectories, which enabled more holistic assessments of motor maturity.

**Results** Recordings were technically successful in all infants, and the algorithmic analysis showed human-equivalent-level accuracy in quantifying the observed postures and movements. The aggregated recordings were used to train an algorithm for predicting a novel neurodevelopmental measure, Baba Infant Motor Score (BIMS). This index estimates maturity of infants' motor abilities, and it correlates very strongly (Pearson's $r = 0.89$, p < 1e-20) to the chronological age of the infant.

**Conclusions** The results show that out-of-hospital assessment of infants' motor ability is possible using a multi-sensor wearable. The algorithmic analysis provides metrics of motility that are transparent, objective, intuitively interpretable, and they link strongly to infants' age. Such a solution could be automated and scaled to a global extent, holding promise for functional benchmarking in individualized patient care or early intervention trials.

## Plain language summary

Assessment of an infant's motor abilities is a key part of regular health checks of infant development. However, there is shortage of methods that would allow objective and user-friendly tracking of infant motor abilities. We describe a system that measures infant's posture and movement with sensors that are attached to the clothing. Movement signals are analyzed with a deep learning algorithm to predict maturity of motor abilities. The accuracy of analysis is comparable to human assessments. This system could enable early diagnosis of developmental delays, and it can be used to assess motor development in clinical trials.

[1] BABA Center, Pediatric Research Center, Department of Clinical Neurophysiology, New Children's Hospital and HUS Imaging, Helsinki University Hospital, Helsinki, Finland. [2] Department of Pediatric Neurology, Children's Hospital, Helsinki University Hospital and University of Helsinki, Helsinki, Finland. [3] Department of Design, Aalto University, Otaniementie 14, FI-02150 Espoo, Finland. [4] Unit of Computing Sciences, Tampere University, P.O. Box 553, FI-33101 Tampere, Finland. [5] Department of Physiology, University of Helsinki, Helsinki, Finland. ✉email: airaksinen.manu@gmail.com; sampsa.vanhatalo@helsinki.fi

Early neurodevelopmental care is globally challenged by a scarcity of objective and scalable solutions available for early neurological assessments[1]. More than one in ten infants require active medical follow-up due to perinatal events or abnormal neurological findings[2]. Only a small minority of these infants will be eventually diagnosed with severe disabilities, such as the severe type of cerebral palsy[3], while a larger portion of infants will develop mild or moderate neurocognitive impairments, such as disorders of communication and attention[4,5]. All of these conditions prompt early therapeutic interventions[6]. However, how to distinguish these infants from the majority, who will show a typical range of neurodevelopmental outcomes despite early concerns, has remained elusive.

The early development of an infant's motor abilities provides an essential framework in the developmental cascade related to language and cognitive abilities[7–12]. This has prompted a wide clinical and research approach to survey or observe how the infant reaches developmental milestones, such as rolling, sitting, or walking[13–15]. They are useful for a wide-scale population screening for developmental delays, and they even generalize fairly well across different cultures[13,15]. However, milestone assessment does not quantitate the spontaneous motor ability of infants, and it is not sensitive to the wide variability that characterizes natural motor development[16,17]. More fine-grained information can be obtained by trained professionals using standardized neurodevelopmental assessments[14,18–20], which collate empirical sets of clinically observable or testable items, such as side turning or holding a toy, which requires an evaluation that is at least partly subjective. These test batteries are performed in a controlled environment, such as a doctor's appointment, which is an unnatural situation from an infant's perspective, compromising ecological validity from the assessor's perspective. There is hence a demand to develop methods for early neurodevelopmental tracking that are robust to variability in infant physiology, the skills of the assessor, and the testing environment[1,19,21]. This could be solved with an objective measurement of spontaneous behavior at home, the most ecologically valid environment. Recent progress in sensor technology has made it possible to record extended periods of infants' spontaneous motor ability in out-of-hospital settings[22–24], with quantitation of behavior at an accuracy that compares with human observers[22].

Here, we set an overall goal to construct and validate a generalizable, scalable, and effective method to measure infants' spontaneous motor ability across all milestone levels of infant motor development from lying supine to fluent independent walking. In the current study, we present an infant wearable that enables widely scalable out-of-hospital studies and recordings of a total of 59 infants during their spontaneous play. We then develop a novel, unified structured scheme for classifying infant postures and movements (hereafter collectively called "motor ability") for each second of recording and test its accuracy and generalizability across infant age groups and human observers. We train a deep-learning-based classifier to mimic human annotations of infant motor ability, which in turn enables the construction of posture- and movement-specific developmental trajectories. Finally, all the wearable data are combined to train an algorithm to predict a novel neurodevelopmental index Baba Infant Motor Score (BIMS), which estimates infants' maturity of motor ability, to be used in individual tracking of neurodevelopment.

## Methods

**Study design**. A primarily cross-sectional cohort of infants was recruited to develop a methodology for quantifying spontaneous motor ability by using a wearable suit, "MAIJU" (Motor ability Assessment of Infants with a JUmpsuit) (Fig. 1a, b). Parts of the sessions were recorded with a synchronized video to allow visual annotation of posture and movements (Fig. 1c) according to a novel infant motor ability description scheme. A self-supervised learning method was employed to confirm that these motor ability classes are genuinely present in the movement signals. Then, a deep-learning-based automatic classifier was trained to analyze infant posture and movement at a second-by-second level in all wearable recordings. These classifiers were shown to perform at a human-equivalent level, enabling the construction of computational indexes for assessing the maturity of infant motor ability (or BABA Infant Motor Score, BIMS), which was compared to a clinically-used assessment scale and parental surveys.

**Participants and recordings**. Infants ($N = 59$) were recruited from the Children's Hospital, Helsinki University Hospital, Helsinki, Finland, to participate in a larger study that assesses neurodevelopment in low-risk term-born infants ($N = 38$) as well as infants with mild perinatal asphyxia ($N = 10$) or prematurity ($N = 11$). Respectively, the recruitment criteria in these three arms were prematurity below 28 weeks gestational age, clinical suspicion or diagnosis of mild perinatal asphyxia in term-born infants, as well as no clinical incidents (low-risk, healthy controls) in term-born infants. For performing MAIJU recordings, we had no exclusion criteria, as the wearable testing and algorithmic development was not expected to be affected by the infant's clinical condition. As all 59 infants were followed-up, 55 were found to develop typically, while four developed a neurodevelopmental condition. The recordings from these four infants were used in the training of the motor ability classifier, but they were excluded from the training of BIMS score, as well as the analyses of age correlations. While this cohort was primarily cross-sectional, five infants were recorded twice with an interval of 6 to 12 months, yielding a recording dataset of $N = 64$ recordings at ages 4.5 to 19.5 months. Corrected age was used for prematurely born infants.

The infants were recorded with the MAIJU wearable (see below) at home ($N = 40$) or they came to a home-like environment (see below) in the BABA center due to logistical convenience ($N = 24$). The infant was dressed in the MAIJU suit, and the recordings lasted for 18 to 199 minutes (average 67 min), with a total recording time in the cohort corresponding to 71 h 30 min. Out of this time, 29 h (18–74 min per infant, average 43 min) in $N = 41$ infants were video recorded to allow motor ability annotation for classifier training (Fig. 1b, c). During the recording, children were allowed and encouraged to move about and play freely, and with minimal disturbance from the adults.

The recording environment was somewhat variable between infants, which might have affected their behavior on top of the situational variance that is naturally present in spontaneous activity. There may be marked differences between homes in terms of physical layout, furniture, or child-relevant objects such as toys. However, a child's own home is still the environment that is best known by the given child, hence it may be considered ecologically relevant for studying natural behavior. Some infants could not be recorded at home for various reasons (e.g., logistics or parent's preference), and they came to our research lab, BABA center (www.babacenter.fi). BABA rooms are relatively large ($4 \times 4$ meters) with a large window for natural lighting, as well as typical household furniture including table, chairs, book chest, carpet, and age-appropriate toys. While this environment is not equal to a child's own home, our experience has shown that it is natural enough to encourage children in a seemingly normal exploratory behavior.

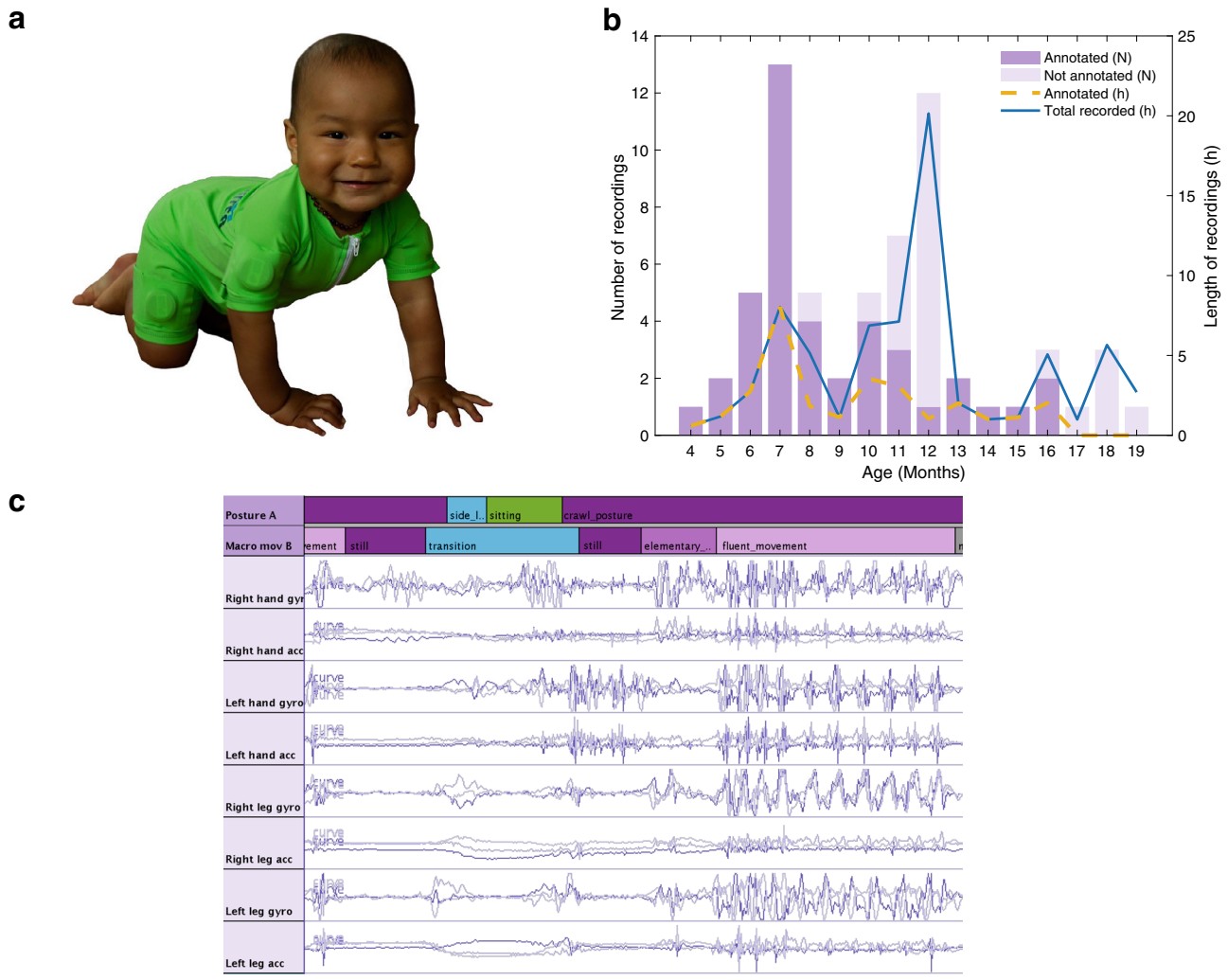

**Fig. 1 Overview of the MAIJU wearable, infant cohort, and recording data. a** A 10-month-old subject crawling at home with the MAIJU jumpsuit, equipped with movement sensors in the proximal pockets of each limb. The photograph has been published with informed parental consent. **b** Summary of the infant cohort ($N = 59$ infants, $N = 64$ recordings) recorded in the present study. Bars depict a monthly breakdown of the numbers of infants with MAIJU recordings with vs without synchronized annotated video recordings, as well as the total length of data available for each age. **c** An example recording in the annotation software showing 20 s of the raw 24-channel data obtained from the four MAIJU sensors, as well as the respective human annotations for postures and movements shown in the bars above the signals, colored according to the motor ability categories shown in Fig. 5a. Note the frequent transitions in posture and movement categories.

**Research governance.** The study was carried out in accordance with the Declaration of Helsinki and good clinical practice guidelines. Ethical approval was obtained from the Ethical Committee of Children's Hospital in Helsinki, the study was approved by the Children's Hospital, and informed written parental consent was obtained for each infant. The study was an observational methodological development study, and thus not registered as a clinical trial.

**Description of the MAIJU wearable.** The novel wearable MAIJU (Fig. 1a) was developed for an unobtrusive and comfortable tracking of spontaneous movement. The MAIJU suit was designed and manufactured in different sizes to fit tightly and comfortably on infants throughout the age range of interest. Little pockets with sensor connectors were laminated proximally on each limb to keep the sensors out of the infant's reach. The garment was designed to tolerate repetitive laundry washing using detergents for synthetic materials. The fabric is akin to those used in swimming suits, i.e., a blend of polyamide and elastane to enable easy movement and a good fit for variable body shapes.

The additional characteristics of the fabric include moisture transportation, stain repellency, quick-drying, and mechanical stability over multiple use cycles. Prior studies[22] have shown that four sensors placed proximally on each limb is enough to provide a reliable estimate of body posture, and the sampling rate of 52 Hz is sufficient for capturing details relevant to infant-typical movement types. The waterproof sensors (Movesense, Suunto, Finland) record tri-axial linear acceleration (accelerometer; m/s2) and angular velocity (gyroscope; deg/s), streaming the data wirelessly via Bluetooth 4.0 or 5.0 low energy (BLE) to an iOS mobile data logger application (Kaasa Solutions GmbH, Düsseldorf, Germany).

**Development of the motor ability description scheme and visual annotations.** We developed a phenomenological motor ability description scheme (Fig. 2) for a comprehensive, transparent, and minimally ambiguous annotation of video recordings during the infants' spontaneous activity. The scheme had to adequately fulfill three constraints: (1) Being descriptive of all time periods of independent movement, (2) being captured by

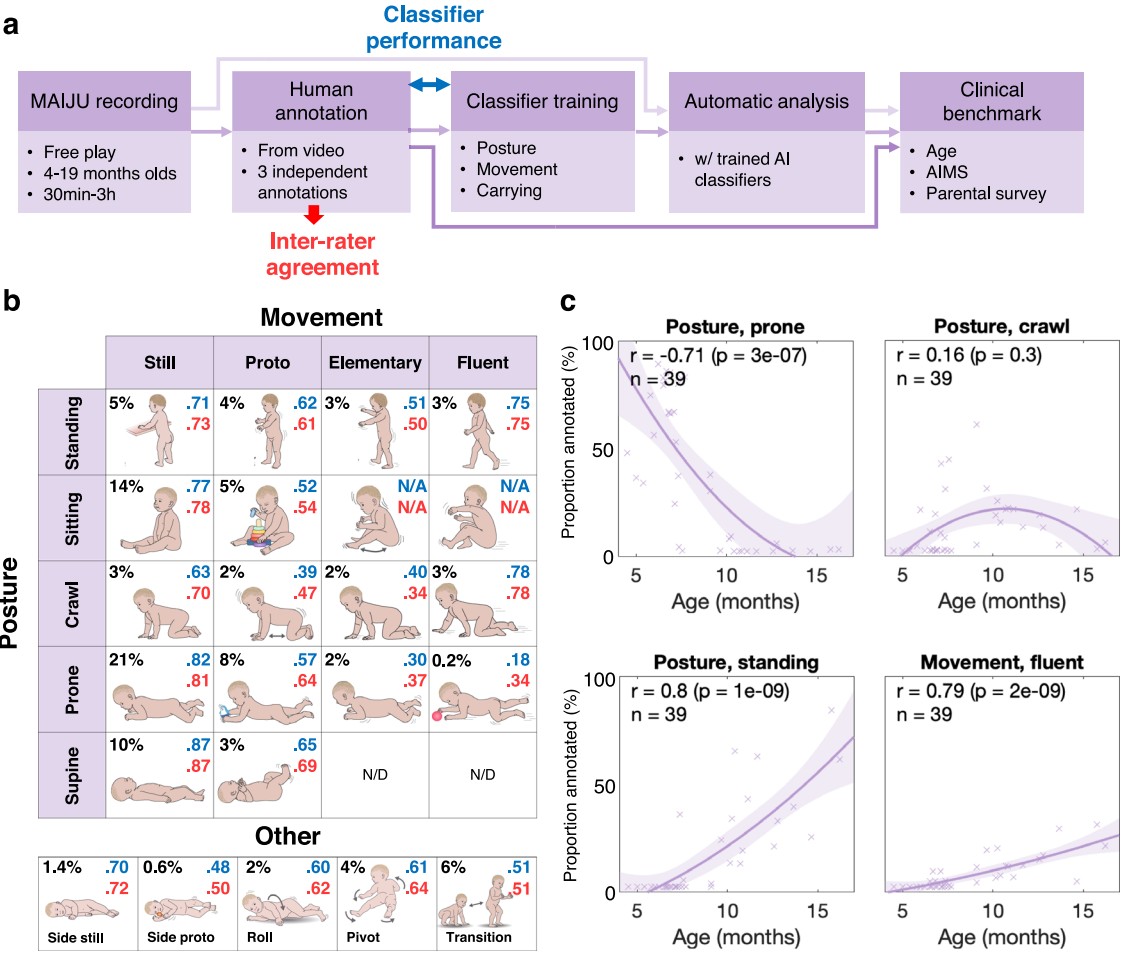

**Fig. 2 Study design and the infant motor ability description scheme. a** Flowchart depicting the overall study design. Coloring of the classification comparisons between humans (red) and human vs algorithm (blue) correspond to the same colors in section B. **b** Illustrations of the posture and movement categories identified in our motor ability description scheme. Numbers in each cell depict the proportion of each category within the annotated dataset (black), and the Fleiss' kappa agreement between human observers (red) or between the algorithmic analysis and human observers (blue) in the classification of 2.3-s signal frames. **c** Correlation between infant age and the proportions of motor ability types ($N = 42$) identified from the video recordings by the human observers (individual points; the line indicates a quadratic regression model with 95% confidence intervals; r represents the Pearson's correlation coefficient). Note a robust age-related decrease in prone posture, increases in standing and fluent movement, as well as the bell-shaped developmentally transient occurrence of the crawling posture.

movement sensors, and (3) retaining an interpretable meaning from visual assessment. The resultant scheme recognizes five different postures and four movement levels in a manner that is physically observable with movement sensors and does not require observers' inferences, such as estimating the child's intention, which is commonly used in the clinical assessment scales[25]. A specific description is given in the supplementary material (Supplementary Tables S1–3). The description scheme was developed through an iterative process[22] with frequent discussions using video examples and test annotations, and comments were invited from external informants to ensure both content and clarity.

Each study with a synchronized video recording was annotated by two ($N = 9$) or three ($N = 32$) independent human annotators ($N = 5$) trained for the task and with a background in infant health care or infant research. The inter-rater agreement was measured by the Fleiss' kappa score computed from the compounded confusion matrices, as well as by confusion matrix-based metrics such as accuracy, recall, and F1 score (see Supplementary Figs. S1, 2). The Fleiss' kappa was used as the primary performance metric, both in overall multi-class performance as well as class-specific classification performance.

**Development of the motor ability classifier**. The motor ability classifier was trained as an end-to-end convolutional neural network (CNN) with a specialized structure that takes in as input pre-processed sensor signals in 2.3-s (120 signal sample) frames with 50% overlap and outputs frame-by-frame categorical probabilities for posture and movement. The preprocessing of the signals consists of the removal of gyroscope bias, linear interpolation of received sensor recording packets into a common ideal time-stamp base (with a sampling frequency of 52 Hz), and temporal smoothing with a seven-tap median filter. The structure of the classifier model was similar to the one used in our previous research[22]. It consists of an encoder module, which produces a frame-specific fused representation of the sensor signals, and a classifier module, which models the frame-to-frame temporal structure of the signals and finally produces the classification output. The posture and movement classifiers were trained separately using the same model architecture. The architecture and implementation details are presented in Fig. 3. The entire annotated dataset ($N = 41$, 29.3 h, 91449 frames) was used for training the system to be used for the classification of unannotated data, and the annotated data was classified with tenfold cross-validation.

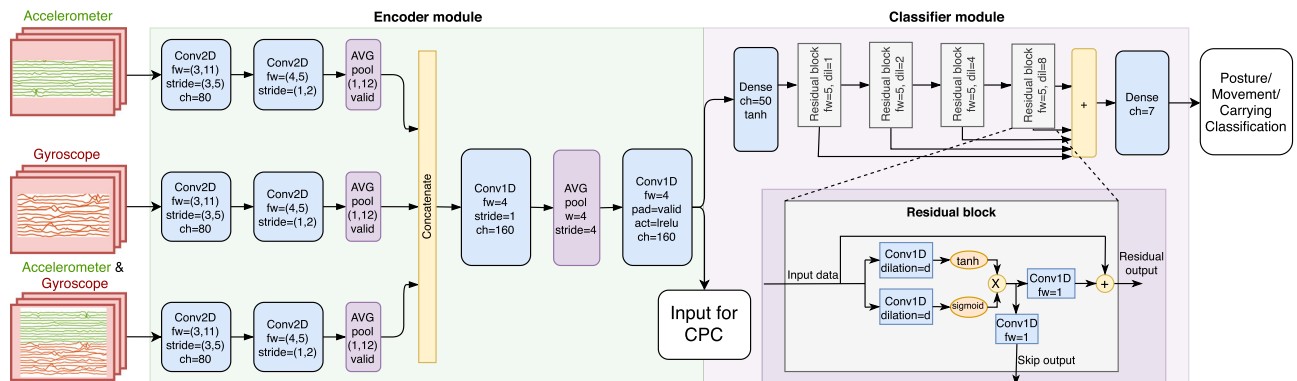

**Fig. 3 Block diagram of the deep-learning-based motor ability classifier architecture.** Abbreviations: activation function (act), average (AVG), channels (ch), convolution operation (conv), dilation (dil), filter size (fw), leaky rectified linear unit (lrelu), padding (pad). The encoder module performs frame-level sensor fusion to obtain a 160-dimensional latent expression of the raw accelerometer and gyroscope signals. The classifier module models the frame-to-frame time dynamics of these features and outputs softmax probabilities for each category separately for each of the classification tracks (posture, movement, and carrying). The training was performed with minibatch gradient descent using the ADAM algorithm (batch size 100 consecutive frames, learning rate $10^{-4}$, beta1 = 0.9, beta2 = 0.999, epsilon = $10^{-8}$) with a weighted categorical cross-entropy loss. In the loss function, each frame's error was weighted with the inverse probability of the target class's occurrence in the training data to mitigate the effects of unbalanced category distributions within the training data. Sample dropout ($p = 0.3$), as well as sensor dropout ($p = 0.3$), was also applied randomly to the input signals during training to ensure the robustness of the trained models. The training was run for 200 epochs and held out validation data (20% of training data) was used to select the best performing model in terms of the unweighted average F1 score. The code for the motor ability classifier was implemented with Tensorflow (v.1.12.0) and Python (v.3.6.9). The code is available at request.

The performance of supervised machine learning classifiers depends on the consistency of the training annotations. Here, we wanted to utilize all available human input, including time instances with a varying agreement between the annotators. The inter-rater ambiguities in the classifier training data were resolved by combining human- and machine-generated labels in a probabilistic fashion using the iterative annotation refinement (IAR) procedure introduced in ref. [22]. In IAR, contested frame annotations (which might suffer from human inconsistency) are weighted with a classifier's probabilistic decision (which can be thought of as being consistent for all samples) to obtain more consistent ground-truth targets for classifier training.

**Analysis of latent signal structures with self-supervised learning.** To obtain a general understanding of the signal structure present in MAIJU recordings, we employed contrastive predictive coding (CPC[26]) to learn a robust latent signal representation based on 42 h of unannotated MAIJU data (Fig. 4a). CPC is a self-supervised machine learning method, which utilizes a learnable encoder model to map the raw signals of an analysis frame at time $t$ into an $n$-dimensional latent representation $Z_t$. In CPC training, the time structure of these latent states is modeled with a recurrent neural network (RNN) model to obtain a time-compounded representation $C_t$ for each analysis frame, capturing the history of the encoded signal up to that point in time. From $C_t$, future states of the encoder latent representations $Z_{t+k}$ are predicted with a simple linear projection. The training is done by applying the InfoNCE loss[26] between the target encoder value $Z_{t+k}$, the predicted value $\hat{Z}_{t+k}$, and a set of contrastive samples $Z_{\neq(t+k)}$ that are negative samples drawn from another section of the recording. As a result of the training, CPC learns an encoder representation that best supports the separation of true future signal states from false potential future signal states, hence capturing structural discriminative properties of the data without data labels. In the present work, the encoder model trained with CPC was identical to the supervised motor ability classifier (Fig. 3). The latent dimension was $n = 128$, the gated recurrent unit (GRU) was used as the RNN model, a prediction distance of $k = 5$ frames (~5.8 s) into the future was used, and the InfoNCE

loss utilized ten negative samples randomly drawn from the same recording. The model was trained for 50 epochs.

**Development of the carrying detection classifier.** Since MAIJU is primarily a wearable method for out-of-hospital recordings, it was essential to minimize the need for active parental input during the recording. The at-home recordings were instructed to contain a designated "playtime" of at least an hour, during which the parents were encouraged to let the infants play independently as much as possible. Since the parents would still be allowed to guide or possibly carry the infant during such playtimes, we found it important to build an additional layer of preprocessing that would automatically detect periods of independent infant movement versus movements due to external forces, such as parental carrying. To this end, we annotated the data and trained an additional frame-level binary classifier for active carrying detection (ACD) to be run at the preprocessing stage before motor ability classification. The ACD dataset consists of a subset of 17 videoed recordings from the full dataset that were performed at infants' homes (total length of 17 h). The annotations for the ACD task were performed with a scheme of five categories: independent movement (i.e., an infant has no contact with anyone), passive support (e.g., infant sits and leans on the parent), active support (e.g., parent supports walking by holding hands), passive carrying (i.e., an infant is being held but no movement is present), and active carrying (i.e., an infant is moved by carrying) (see Supplementary Figs. S3, 4). The deep-learning classifier structure was identical to the motor ability classifiers (cf. Fig. 3). The most reliable detection performance (leave-one-subject-out a.k.a. LOSO cross-validation) was achieved with binary classification for active carrying (97.2% accuracy; 54.5% recall, 58.1% precision for carrying; 98.7% recall, 98.5% precision for non-carrying), which means that roughly half of the frames with carrying can be automatically filtered out from further analysis at the expense of only very few false detections. The trained (and cross-validated where applicable) ACD classifier was applied to all analyses on MAIJU recording distributions (Supplementary Fig. S5), which means that these results are obtained with a realistic use-case scenario.

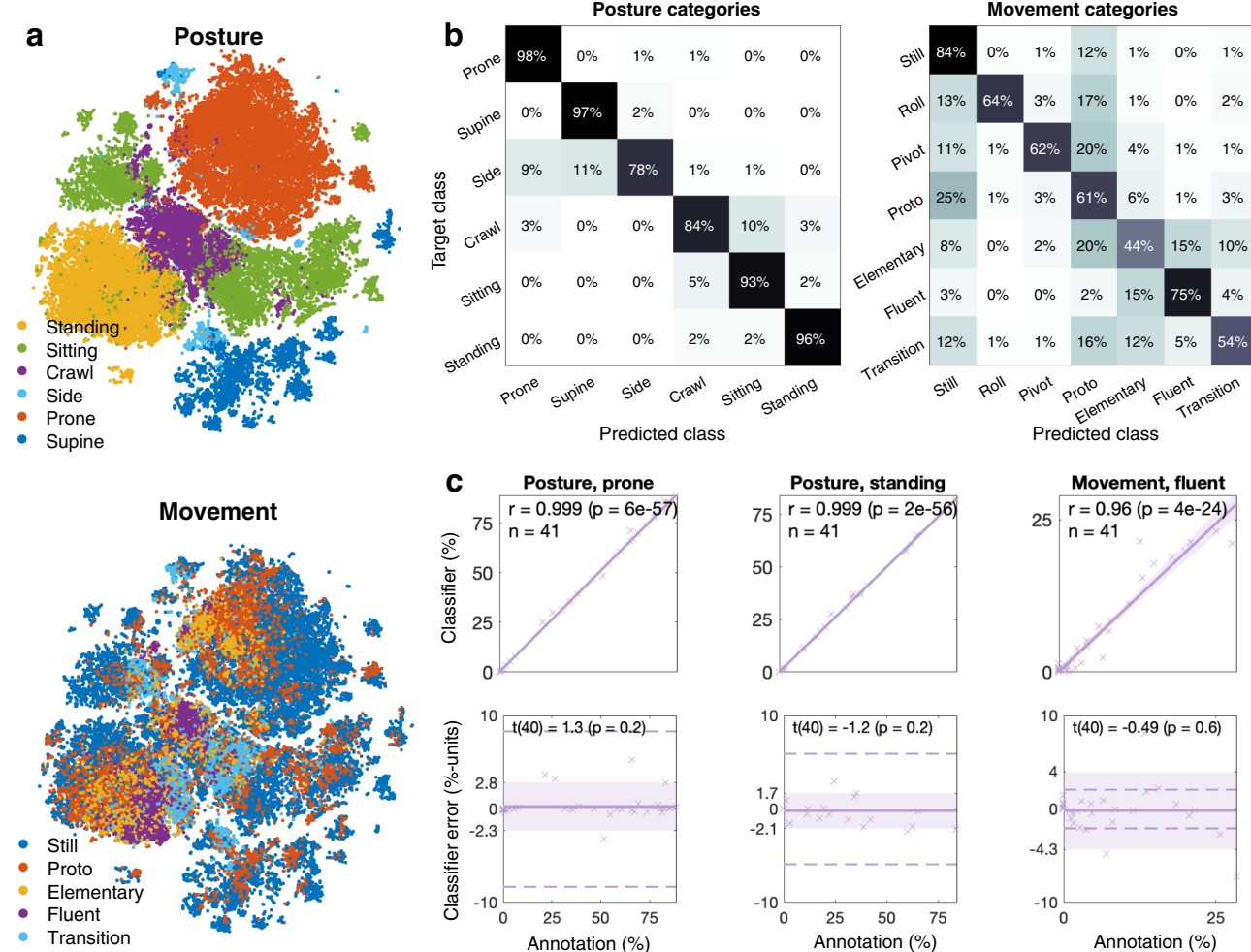

**Fig. 4 Classifier development and computational analyses of MAIJU recordings. a** t-SNE plots obtained from self-supervised feature embeddings (CPC) with color codings for posture (top) and movement (bottom). Note the clear clustering of posture categories, while the movement categories show relatively more dispersion. **b** Confusion matrices showing recall values (in %) of the algorithm output ("Predicted class") and the compounded human expert annotations ("Target class"). Note the high numbers in the diagonal line indicating high agreement. **c** Comparison of quantified motor ability between the classifier and human annotations (N = 42). In the upper graphs, the scatter plots show the proportion of time spent in the given postures or movements as estimated by the classifier algorithm (Y-axis) and the human annotations (X-axis). The Pearson's r (and its p value) denotes the linear correlation between the proportion values. Below, the Bland–Altman plots of annotations vs classification errors are shown for assessing whether the classification errors have a systematic bias and/or are dependent on the amount of posture/movement identified by the algorithm. The stippled lines depict an average one-month developmental change (percentage points per month) as taken from a linear regression model fitted between the age (in months) and the given motor ability occurrence (cf. Fig. 2c). Note that 100 and 88% of the measurements in posture and fluent movement categories, respectively, are within these stippled lines. The shaded zone depicts the 95% confidence interval (in percentage points) of the classifier error. The t value depicts the two-tailed t-test result (with 40 degrees of freedom) on the null hypothesis that the error has a mean of zero; this shows that the proportional estimates are unbiased.

**Performance assessment of the motor ability detection algorithm.** The performance of the motor ability classifier was tested at multiple levels. At the lowest level, the performance of the frame-to-frame classification was measured based on a compounded confusion matrix: Recording-level tenfold cross-validation was utilized to produce test-set predictions from the left-out recordings. Similar to the inter-rater agreement analysis, the predictions from all folds were compounded against all of the original human annotations into a single (posture- or movement-specific) confusion matrix (Fig. 4b and Supplementary Fig. S6). Additionally, the compounded confusion matrices against the IAR-derived training targets are presented in Supplementary Fig. S7.

The practically most relevant performance measure is the accuracy of the recording-level motor ability distributions, which

provides the primary output to be used in subsequent analyses. Notably, if the errors on the short-term signal frame classification are unbiased, they will average out with sufficiently long recordings. Hence, recording-level distributions combined with recording length analysis are the most informative for estimating the overall feasibility of the method. We measured this performance by a two-stage analysis for each annotated category (Fig. 4c; full set in Supplementary Figs. S8, 9): first by measuring the correlation of the annotated and classifier-produced category distributions (as measured by Pearson's r and its p value), and second, by the Bland–Altman plot analysis between the annotated distributions and the classifier error. The two-tailed t-test is used to test for the null hypothesis (at p < 0.05) that the errors have a mean of zero. To add further context to the Bland–Altman analysis, we estimated the standard deviance of the error (±2 SD

error area, colored) and we also compared the error to a monthly age-related change Δ (as percentage points per month of age; drawn with dashed lines). The Δ-values were computed as the slope of a least-squares linear regression model fitted into the [age, distribution value] scatter pairs from the annotated dataset. Notably, this representation is informative only for categories with monotonical age-dependent distributions (e.g., standing and fluent movement).

**Development of the BABA infant motor score (BIMS) metric.** The BABA infant motor score (BIMS) predictor was designed to predict the relative maturity of an infant's motor ability, which in the typical infants reflects the most likely age of the infant based on the classifier-produced category distributions of MAIJU recordings (see Supplementary Fig. S10): the posture and posture-conditional movement distributions. In the classifier, age-dependent multivariate Gaussian distribution models of the MAIJU motor ability distributions were estimated from the dataset of typically developing infants ($N = 55$ infants; $N = 60$ recordings) using an age resolution of 1 month, and including recordings within ±1 month from the center age in the estimation process. After the estimation, the BIMS of a new unseen recording was computed by first computing multivariate Gaussian likelihood for all of the age-dependent models, given the motor ability distributions from the present recording, followed by calculation of the weighted average of the ages corresponding to the Gaussian models using the relative likelihoods of the models as weights. The evaluation of the method was performed with LOSO cross-validation to produce age estimates for all of the held-out subjects, after which Pearson correlation between the target and predicted ages was used as the performance metric. Due to the limited availability of data for all age groups, diagonal covariance matrices were used in the multivariate Gaussian models. At least 3 recordings were used to determine the means and standard deviations of each age bin. The standard deviations were set to have a minimum value of $10^{-4}$ to ensure model stability. If at least three recordings were not found in the ±1 month range from the center age, recordings with the smallest age difference to the center bin were added into the group until 3 recordings were obtained. The modeled age ranges were from 4 months to 16 months, where the 16 months age pool included all recorded children who were over 16 months of age. This was motivated by the fact (see also Fig. 4a) that infant motor ability in our description scheme saturates at around this age[27,28], just like the well-known ceiling effect of AIMS values after 18 months of age[27]. Likewise, the target age within BIMS prediction evaluation for children over 16 months was set to 16, and the oldest age group was labeled as "16+" in Fig. 4b. Similar logic would also apply for infants younger than 4 months (not present in the dataset), which makes BIMS a bounded scale of continuous values (4 to 16 months) that are normalized into a scale of [0–100] with BIMS = (predicted_age_in_months$-4$) $\times$ 100/(16$-4$), where 0 denotes "non-mature" motor ability (as in the ≤4-month-olds' group), and 100 denotes "fully matured motor ability" (as in the ≥16-month-olds' group).

We tested the robustness of the BIMS estimates in relation to the length of the recording time from which the MAIJU distributions are computed. To reach this end, we systematically measured the mean absolute error (MAE) on the BIMS-classifier (with LOSO cross-validation) with recordings of varying lengths. The classifier was trained with the full dataset (same as in the main BIMS experiment), but for testing, we utilized only the recordings with usable length over 120 min ($N = 12$) to ensure the underlying uniformity of the test data. From these recordings, we sampled subsegments (with random start times) ranging from

10 to 100 min at 10-min intervals and computed the BIMS age based on each segment's distribution. The sampling of the segments was performed for 1000 iterations, and finally, the MAE between the BIMS scores and the true ages were computed. The MAE variability as a function of recording segment length is visualized with a boxplot where the median, interquartile range, and range of the data distributions are shown.

**Comparison of the algorithmic output to clinical development.** We devised a visualization approach (Fig. 5a) to provide an intuitive and easy-to-interpret picture of the MAIJU-classifier derived distributions that capture developmental change in infant motor ability as a function of age. Within the visualization, the age-category pooled (same as in the BIMS classifier; center bins 1 month apart, pooled with recordings from ±1 months) averages of category distribution values are plotted with a violin plot to highlight their deviance from zero. For the sake of visual clarity, the left/right categories have been fused for both the movement and posture tracks, and for the movement track, only a selected number of posture-dependent movement categories are shown, which highlight the development of the typical movement modalities: prone crawling, crawling, and walking.

A subset ($N = 28$, age range 8–17 months) of the recorded dataset was clinically evaluated by an experienced physiotherapist (T.H.) according to the Alberta Infant Motor Scale (AIMS[27]) on the same day as the MAIJU recordings. The Pearson correlations ($r_{AGE}$, $r_{BIMS}$) between the raw (not age-adjusted) AIMS score and infant age (chronological age and BIMS) were measured. To test the hypothesis that the BIMS classifier corrects infants' ages in the direction of their motor developmental level, we utilized the two-tailed comparing correlations (coror[29]) test between $r_{AGE}$ and $r_{BIMS}$.

Finally, another subset ($N = 20$) of the infant cohort had an additional parental survey collected to evaluate the parents' assessment of the amount of time the child typically spends in different postures. We used a larger custom-made questionnaire to assess many aspects of the project, including MAIJU design, infant development, and parents' perception of various things. This questionnaire was delivered on paper and it was requested to be filled by the parents/caregivers at the time of MAIJU recording. For the present study, we chose two questions to be compared with MAIJU outputs: estimate the average amount of infant's free playing time spent in (1) crawl posture and (2) sitting posture. The answers were given on a verbally explained scale (in Finnish), whose scale ranged from 1 to 9 (*never, very rarely, rarely, sometimes, often, very often, most of the time*). The survey answers were compared to the MAIJU-derived posture distribution values using Spearman's rho (Fig. 4e) to estimate the reliability of such quantitative assessment. As the MAIJU-derived posture distributions can be assumed to be very close to the ground truth for the given recording session, discrepancies between the recorded distributions and survey answers can be mainly attributed to two sources: (1) the normal day-to-day variability of infant movements, and (2) estimation error of the parents. Our setup does not allow differentiating the relative sizes of these effects, but valuable intuitive insights can be gained by comparing the results between multiple posture categories.

**Statistics and reproducibility.** Data preprocessing and analysis were performed using custom Matlab codes (version 2021a). The analysis codes are publicly available at Zenodo[30]. The raw figure data have been made available in Supplementary Data S1.

Due to the large class frequency imbalance expected for the recordings (e.g., older infants rarely crawl, whereas younger infants do not stand), classification performance and inter-rater

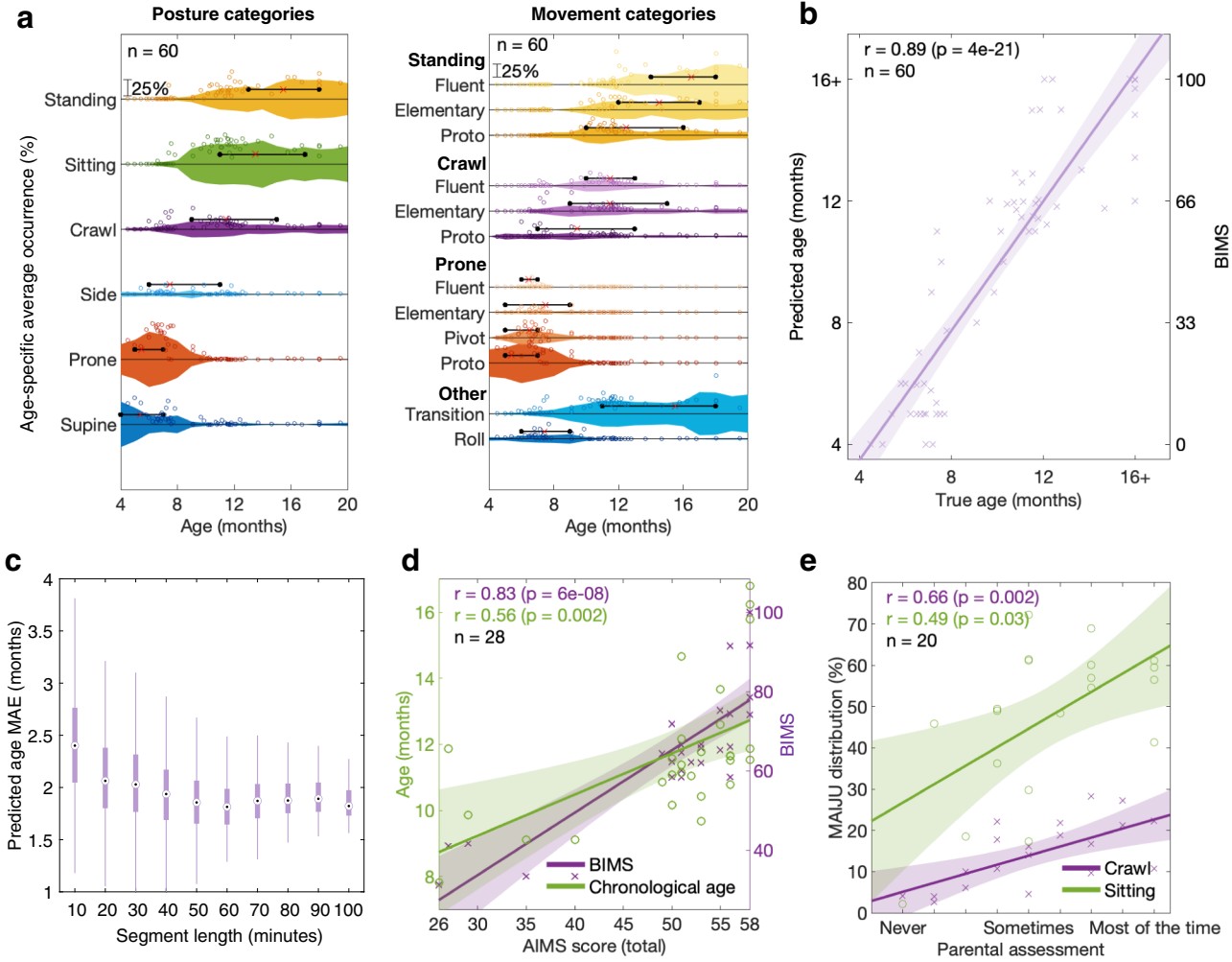

**Fig. 5 Assessing maturation of infant motor ability with MAIJU. a** Graphs showing the occurrence of each posture (left) and motor ability class (right) as a function of infants' prematurity-corrected age ($N = 60$). The black lines denote the interquartile range (IQR) of the age-related occurrence, and the red cross depicts the median age for the occurrence. The measures combine all analyzed 2.3 s time frames of the recording session, and all infants exhibit motor ability in several classes, which show clear developmental trajectories. Note also the clear developmental sequence in the movement categories within each posture. **b** Scatter plot showing a correlation between infants' ($N = 60$) chronological age and the age prediction from the BIMS algorithm. **c** Dependence of BIMS estimate on the length of recordings between 10 and 100 min of data. Data were taken as randomly sampled segments from $N = 12$ recordings whose length was over 120 min (range 121–150 min). The findings in the Y-axis are expressed as the mean absolute error (MAE) in the age prediction as in **b**) (bars show the median, IQR, and the range). Note how the MAE stabilizes with recording lengths over one hour. **d** Correlation between BIMS and AIMS score (purple) compared to the correlation between true age and AIMS score (green) ($N = 28$). The result indicates that the BIMS score is biased towards the actual developmental level, as the correlation is significantly higher (cocor tests; $p < 0.05$; $N = 28$) compared to the chronological age correlation. **e** Comparison between a parental estimate of infant's time spent in various postures and the MAIJU-derived corresponding measures ($N = 20$).

agreement were measured using a compounded confusion matrix. Each individual recording's (posture or movement-specific) confusions were summed into a total confusion matrix for all data, from which relevant statistics were computed (kappa, F1, accuracy).

Standard tenfold cross-validation was utilized for classifier evaluation, where the individual recordings were split into ten equal-sized groups (folds) at the recording (participant) level, from which nine folds were used to train a classifier while testing on the remaining unseen data fold. The process was then repeated for each possible test fold. Within the training data, 80% of the frames were used for classifier training and 20% for validation. The training was stopped when the classifier had reached maximum performance for the unseen validation data based on the unweighted average F1 score.

Correlations were computed using Pearson's $r$, where $p$ values were computed using the two-tailed null hypothesis that the correlation is zero, except for the parental questionnaire data where standard Spearman's rho was applied with the same two-tailed null hypothesis. In Bland–Altman analysis, the two-tailed $t$-test was used to test the null hypothesis (at $p < 0.05$) that the errors have a mean of zero.

The comparing correlations (cocor[29]) test battery was used to study the statistical significance of different correlation values with the following settings: two dependent groups, overlapping correlations, null hypothesis: r.jk = r.jh, alternative hypothesis: r.jk ≠ r.jh, alpha level 0.05. The cocor test includes the following ten sub-tests: (1) Pearson and Filon's z, (2) Hotelling's t, (3) Williams' t, (4) Olkin's z, (5) Dunn and Clark's z, (6) Hendrickson, Stanley, and Hills' modification of Williams' t,

(7) Steiger's modification of Dunn and Clark's z using average correlations, (8) Meng, Rosenthal, and Rubin's z, (9) Hittner, May, and Silver's modification of Dunn and Clark's z using a backtransformed average Fisher's Z procedure, and (10) Zou's confidence interval. No outlier or other exclusion criteria were applied to the data, as we assumed all data to be representative of data captured in and outside the lab.

**Sample size calculation**. This study was observational, not interventional, hence sample size was not calculated.

**Reporting summary**. Further information on research design is available in the Nature Research Reporting Summary linked to this article.

## Results

**Design of and recording with the infant wearable**. For a reliable assessment of infant motor ability, we designed a wearable solution MAIJU (Motor ability Assessment of Infants with a Jumpsuit; Fig. 1a) by equipping a garment commonly used as a swimming suit in many cultures with sensors. Altogether 59 infants, with an age range of 4.5 to 19.5 months, participated in 64 recording sessions (Fig. 1b), performed either at home ($N = 40$) or in the research facility ($N = 24$). All recording sessions included a spontaneous play that could be encouraged with minimal active physical contact by an adult. Part of the cohort (total $N = 41$ infants; 29.3 h; both at home and in the lab) were videotaped to allow visual annotation of the infants' posture and movement (Fig. 1c) for later training and validation of the automated classifier algorithms and to measure inter-annotator consistency for the novel annotation protocol (Fig. 2a).

**Development of a unified, structured scheme for infant motor ability**. Leveraging MAIJU's full potential in motor ability assessment calls for motor ability descriptors that strike a balance between (1) having high temporal accuracy for moments of independent movement, (2) being captured by movement sensors, while also (3) retaining an interpretable general meaning that is visible from visual assessment. Aiming to optimize for these three constraints, we developed a novel motor ability description scheme for recognizing two parallel tracks that together are able to comprehensively describe infants' motor ability (Fig. 2b): five different postures (lying supine, lying prone, crawling, sitting, and standing) and four different movement qualities within them (still, proto, elementary, and fluent). We also included two intermediate postures (lying on the left or right side) and five intermediate movement types (pivoting left/right, rolling left/right, and transitioning between postures). This scheme perceives motor ability via a *posture* state during which the infant will exhibit a graded quality of *movement*, encompassing stillness, general activity ("proto"), developing movement patterns ("elementary"), and mature movement patterns ("fluent"). The detailed descriptions behind the motor ability description scheme are presented in the supplementary material (Supplementary Tables 1–3). Five researchers were trained to independently annotate these motor ability types in videotapes of MAIJU recording sessions ($N = 41$ infants, total duration 1758 min), in order to provide a reliable benchmark for supervised classifier training, as well as to assess human inter-rater agreement (Fig. 2a, b).

A comparison of the second-level posture annotations showed a very high overall inter-rater agreement for all posture types (Supplementary Figs. S1, 2; Fleiss' kappa k = 0.95). The clear-cut postures reached a nearly perfect agreement (supine k = 0.97, prone k = 0.97, standing k = 0.98, sitting k = 0.95), while the

other postures exhibited somewhat lower levels of agreement (crawl k = 0.88; side-lying k = 0.79). Comparison of the movement annotations at a second-level timescale showed an overall inter-rater agreement of k = 0.60) However, it varied widely between different movement types, ranging from substantial agreement with recognizing still (k = 0.67) and fluent movement (k = 0.71) to far lower agreement in recognizing transitions (k = 0.51) or elementary movements (k = 0.4). Note that for chance-level agreement, k = 0. Though unideal, an inter-rater agreement around k = 0.6 is typically described as "moderate" (k < 0.6) or "substantial" (k > 0.6)[31]. Such agreement rates are common in annotation tasks with naturally ambiguous categories, such as various EEG tasks[32] or general movements assessment[33]. The inter-rater confusion matrix for movement (Supplementary Fig. S1b) shows that the confusions between the categories occur between conceptually related motor ability types: e.g., fluent movement is primarily confused with elementary movement, but not with proto or still. The detailed metrics of agreements between human raters and algorithms for posture-conditional movement categories are presented in Supplementary Fig. S1c.

We then tested how the novel motor ability descriptors reflect developmental change over the age range from 4.5 to 16.2 months (based on the full annotated dataset). All metrics of posture and most metrics of movement were found to exhibit a strong correlation to infant age (Fig. 2c and Supplementary Fig. S11). For instance, there was a strong, expected developmental decline in lying prone (Pearson's $r = -0.71$, $p < 0.001$, $N = 42$) and being still (rho $= -0.62$, $p < 0.001$), while standing posture ($r = 0.8$, $p < 0.001$) and fluent movement ($r = 0.79$, $p < 0.001$) showed an expected strong developmental increase. Meanwhile, crawl posture ($r = 0.16$, $p = 0.3$) and transitions ($r = 0.68$, $p < 0.001$) showed a nonlinear U-shaped transient hump peaking around the end of the first year. Finally, these motor ability descriptors correlated strongly to the internationally well-known motor assessment Alberta Infant Motor Scale, AIMS[15] (Supplementary Fig. S12); there was a clear negative correlation with proto movement ($r = -0.71$, $p < 0.01$, $N = 13$) and a strong positive correlation with standing posture ($r = 0.8$, $p < 0.001$) and fluent movement ($r = 0.59$, $p = 0.03$).

**Confirming the presence of motor ability categories in wearable signals with self-supervised learning**. Devising an automated motor ability analysis is critically dependent on the genuine presence of the target motor ability categories in the recorded wearable data. To evaluate this, we used a self-supervised learning method (contrastive predictive coding, CPC[26]) for robust learning of latent representations from the recorded signals[26,34]. A two-dimensional view[35] (Fig. 4a) of the results reveal clear clusters in the data, which match very closely to the posture categories identified independently by the human observers from the corresponding video recordings. In contrast, the different movement qualities exhibit less clear category boundaries. These observations are fully in line with the high human inter-rater agreement for posture, as well as the relative inherent ambiguity in the movement categories, also seen as a lower inter-rater agreement. Taken together these findings suggest that the phenomenologically identified, intuitively, and clinically reasoned motor ability classes (Fig. 2b) are genuinely present in the signals recorded with the MAIJU wearable.

**Construction of a deep-learning-based classifier for infant motor ability assessment**. Comparison of the motor ability classifier outputs to human annotations showed a very high overall agreement for both posture and movement (Fig. 4b and Supplementary Fig. S6) that were on a par with the inter-rater

agreement (Fig. 2b and Supplementary Figs. S1, 2). The average kappa between algorithmic and human annotations for the posture categories was 0.93 (class-specific; e.g., prone k = 0.97; supine k = 0.97; standing k = 0.85), while there was only a modest and expected confusion between nearby postures, such as side versus prone or supine, as well as sitting versus crawling. Confusion matrices between movement categories (combined for all postures) indicate a substantial agreement between algorithm and human for the phenomenologically most distinct categories of still (k = 0.68), rolling (k = 0.62), pivot (k = 0.61), and fluent movement (k = 0.73).

The performance metrics from classifier-to-human vs. human-to-human (Fig. 2b and Supplementary Figs. S1, 2) comparisons suggest that a human annotator can be replaced by the classifier algorithm without notable loss of agreement; this indicates a human-equivalent level of performance for both posture and movement classifications.

Finally, we assessed how accurately the algorithm is able to provide individual-level quantitation of the time spent in different postures and movements normalized by the total recording time (see Supplementary Fig. S5 for procedural details). We found a very high correlation between human annotation and classifier outputs (Fig. 4c; full list in Supplementary Figs. S8, 9) for the fractional time that the infant spends in prone posture ($r = 0.999$, $p < 0.001$), standing posture ($r = 0.999$, $p < 0.001$), and fluent movement ($r = 0.96$, $p < 0.001$). A Bland–Altman analysis indicates that the correspondence between algorithm- and human-based motor ability classification is not significantly biased (two-tailed $t$-test; $p > 0.05$) by the amount of time in a given posture or movement of each individual. Moreover, we compared the classification errors to the population-level change in motor ability distributions per one month of development (obtained with a linear fit to the annotated data) (Fig. 2c). For all infants, the error was below the one-month developmental change for the prone and standing postures. The fluent movement was also quantified within one-month bounds in 88% of infants.

**Assessing infant motor development with the BABA infant motor score (BIMS).** Pediatric developmental assessments are always challenged by the substantial inter-individual variability in both the rate and shape of developmental trajectories[13,14,16,17,20,21]. A child may also spontaneously depart from the expected milestone pathway and adopt alternative strategies in movement repertoire, such as bottom shuffling until walking onset[7,16,17]. Therefore, an experienced child neurologist combines all motor ability patterns of an infant into a holistic, clinical assessment of neurodevelopmental maturity.

Here, we studied whether a global maturity of motor ability can be obtained from the MAIJU data. All posture and movement patterns were found to correlate strongly with the child's age at the recording time (Fig. 2c and Supplementary Fig. S11; age corrected for possible prematurity), and the data pooled over all typically developing infants (N = 56 infants; N = 60 recordings) confirmed that our motor ability metrics comply with the sequence of motor milestones[1,13,36]. Moreover, there was a strong developmental succession from supine to standing postures, as well as from still to fluent movement patterns within each posture category (Fig. 5a). These age-related changes in the motor ability distributions support the training of a probabilistic prediction algorithm that provides a maximum likelihood estimate of a child's age from the MAIJU data distributions (Supplementary Fig. S10). A transparent interpretation of such classifier output is "motor ability age" in months, which compares directly with the everyday clinicians' aim to benchmark a child's motor ability with age-typical performance[36]. We found a very strong correlation

(Pearson's $r = 0.89$, $p < 1e-20$; Fig. 5b) between infants' chronological age and the cross-validated algorithm-generated age prediction, with the mean absolute error (MAE) of 1.4 (1.1) months (IQR range −1.2–0.9, median −0.3) indicating that our quantified motor ability measures follow infant chronological age with striking accuracy. We also assessed how much the prediction accuracy is affected by the recording length, which is a practical challenge in all infant studies. For a subset of the recordings that exceed 120 min in length (N = 12), the average MAE decreases from 2.4 (0.5) months to about 1.9 (0.2) months when the length of MAIJU recordings increases from 10 to 100 min, respectively. A reasonably stable result was obtained when the recording length exceeded 1 h (Fig. 5c).

Next, we wanted to render the motor ability age prediction to a clinically more suitable metric that generalizes across user cases by providing an age-free estimate of infants' maturity of motor ability. This was done by re-scaling the motor ability age prediction to a continuous scale bounded between [0, 100] so that the lowest and highest values represent the least and the most advanced performance, respectively. This re-scaled score was called BABA Infant Motor Score (BIMS), with the range bounded from the youngest (i.e., 4.5 mos) to the oldest (over 16 mos) infants in our cohort. The clinical value of a novel measure like BIMS is measured by its ability to provide added informational value in a clinical context. To estimate such potential, we evaluated the nature of prediction errors in the BIMS estimates and we hypothesized that the error in an infant's age prediction is linked to the actual motor maturity of the given infant. We took the Alberta Infant Motor Scale (AIMS) score as an established metric of a child's motor maturity[27], and we compared the correlation between infants' chronological age and AIMS score to the correlation between BIMS and AIMS (cocor[29]) (Fig. 5d). The correlation between BIMS and AIMS was significantly higher than the correlation between chronological age and AIMS (all two-tailed cocor tests; N = 28; $p < 0.05$), increasing from Pearson's $r = 0.56$ to 0.83 (Spearman's rho = 0.6 to 0.82), which indicates that BIMS discloses genuine diagnostic information in that it is more closely linked to infants' actual motor maturity than to estimating their chronological age.

**Example uses of the MAIJU wearable in the clinical assessment.** Quantified motor ability assessment can be used in diverse contexts. Here, we compared the potential value of algorithmically assessed motor ability maturity in some common clinical and clinical research scenarios. First, we compared the quantified MAIJU category distributions to the widely-used AIMS score[27]. We found expected, robust correlations between multiple MAIJU-derived measures (Supplementary Fig. S12). Akin to the BIMS-classifier, we trained an AIMS predictor that showed a very high correlation with the actual AIMS score (Supplementary Fig. S13; Pearson's $r = 0.93$, $p < 0.001$).

Second, we compared the MAIJU-derived quantitative motor ability measures with information from parent surveys, the key source of clinical information[37,38]. The parents were asked to estimate the amount of time that the child spends in different postures. While there was an expected overall correlation between the by-nature subjective parental estimates and the objective MAIJU-derived results, there was also a salient scatter in many parents' observations (Fig. 5e). The parental surveys assessing quantitative information may be readily confounded by the highly variable motor ability of infants[39]. This was also clearly seen in the MAIJU recording as very high rates of transitions in both posture (average 3.6/min, IQR 1.5–5.2) and movement (average 13.5/min, IQR 10.7–16.1), which both also exhibited a significant correlation to infant age (Supplementary Fig. S14).

## Discussion

We constructed and validated a potentially clinically applicable method for accurate, objective, and quantitative tracking of an infant's gross motor performance throughout the full developmental sequence from lying supine to fluent walking. Our work first conceived a novel description scheme for a transparent and intuitively interpretable classification of an infant's postures and movements, collectively called motor ability. This scheme was shown to correlate with the infant's motor maturation, and it reaches acceptable inter-rater agreement to serve as a reference for training computational classifiers. We then developed a novel infant wearable, MAIJU, for recording an infant's spontaneous movements within both in- and out-of-hospital environments. A self-supervised learning method[26] was employed to verify that these motor ability categories are genuinely present in the infant's movement recording data, which forms a strong foundation for training deep-learning-based classifiers to analyze the MAIJU recordings. The classifier algorithms were shown to perform at the accuracy of human observers. Finally, quantified motor ability metrics correlated strongly with the established sequence of developmental milestones, which supported the training of an algorithmic predictor of age in typically developing infants. This predictor yields a novel, transparent summary measure of motor development, BIMS, which holds promise as an objective universal metric in early neurodevelopmental assessment, supporting individualized neurodevelopmental care as well as benchmarking of clinical trials[40–42].

The potential utility of a reliable, objective, and quantitative method for tracking infant neuromotor development is considerable. Prior work has shown that movement sensors can be used to follow a subject's posture[22,43,44] or motor activity in health and disease, with mild to modest statistical correlation to salient clinical conditions[23,45–47]. However, most of the prior work has been constrained by practical limitations in the recording configuration and data analytics, such as using only one or two accelerometer sensors to quantify only the gross amounts of movements[23,43,48,49]. Here we show that even more sensitive recognition of different motor ability types in the child's spontaneous behavior[16,17,36] may be achieved by combining modern deep-learning signal analysis methods with multisensory recordings. Moreover, the computational analysis is substantially boosted when the classifier training is founded on a well-structured, physiologically reasoned motor ability description to enable second-by-second activity recognition by both human observers and the sensor signals. Our findings back up the everyday observation that infants' spontaneous behavior is characterized by very frequent transitions that jointly provide crucial diagnostic clinical information[17,50,51], hence calling for high temporal resolution in both data capture and analyses.

An unambiguous phenomenological description of infant motor ability is crucial for building objective quantification scales. Several assessment scales[14,15,18] have been developed and are widely used for clinical assessment purposes. They typically consist of a larger set of categorical items, except for the detection of fidgety movements in the context of general movement assessments during the first postnatal months[52,53]. The assessment items could be reported by the parents or observed by a health care professional from video recordings or during a lab appointment. The outcomes of such scales are usually summed scores compacted from multidimensional observations of an infant's overall behavior, which typically include the viewer's interpretations of the infant's intentions and/or other subjective and qualitative items. Notably, none of the existing assessment scales is directly suitable for characterizing infants' motor ability at a high enough temporal accuracy needed to capture the rapid motor ability transitions of a naturally behaving infant[17,36,51].

Our work conceived a template for motor ability description that shows good to excellent agreement between human raters, while it also allows direct translation from the MAIJU recordings to deep-learning-based analytics. The visual and automatic detections of these motilities in an infant's behavior are *per se* interpretable and meaningful for a human observer, and they also link strongly to both the infant's age and the clinical assessment scale AIMS. Yet, it is important to note that this motor ability description scheme was designed with three constraints: it had to support a comprehensive classification of each 2-s epoch of infant's movement into visually recognizable categories that could, at least in theory, be extractable from movement sensor data, and which would be easily interpretable by humans (e.g., clinicians) in order to build trust on any "overall measure of motility" (such as BIMS) by grounding the measure to real-world observable motor phenomena. Our present motor ability descriptors are therefore not informative for attempts to understand other qualities of an infant's behavior, such as intentionality or fine manual operations characteristic of a child's exploratory behavior.

It has recently become popular to directly train deep-learning-based classifier algorithms to turn raw signals into high-level categorical outputs, such as diagnostic[54,55] or clinical outcomes[41,56,57]. A direct clinical diagnostic prediction from the raw data could have been used in our context as well. However, several issues argue against such a strategy. First, a direct prediction from the sensor data would need datasets that are orders of magnitude larger to accommodate the very high complexity in the raw sensor data. Second, a direct prediction would ignore the intermediate and *per se* interpretable result, the detailed class-wise motor ability quantitation, thereby greatly limiting the transparency and flexibility of the approach. For instance, it is easy to envision a wide potential for metrics of individual motor ability classes as independent biomarkers in neurodevelopmental assessment. Movement recognition in the context of posture also provides a straightforward front-end to building further context-dependent movement analyses, such as assessing asymmetry in crawling or walking[16,18,58], which are key aspects in predicting the development of the common condition, unilateral cerebral palsy[59]. Moreover, reliable tracking of postures, such as tummy time, may be essential for studying developmental correlates of infant behavior[60].

A child's age is the most important benchmark in all pediatric assessments, and both structural and functional developments are expected to follow predictable trajectories, the basis of established growth charts[61]. With the development of advanced machine learning-based methods, several novel computational indices have been proposed to predict age from structural, functional, and molecular measures[62–66]. Our study shows that the prediction of an infant's age from the MAIJU data is accurate until it saturates by about 16 months when the typically developing infants reach the upper boundary of our motor ability description scheme[13,21,36,67]. This limits the conceptual utility of direct age prediction, which is overcome by converting the output into the novel BIMS metric. Since BIMS is tightly linked to the age of healthy infants, it readily supports continual learning and validation[68] from new infant cohorts as needed. Moreover, such a unidirectional metric allows direct use of an infant's motor ability tracking in statistical assessments with any other scalar measures from the clinical or other sources. For instance, BIMS provides a straightforward measure for the challenging but topical studies on the developmental cross-domain interactions, such as is seen between infant neuromotor development, later cognitive outcomes, and/or environmental enrichment interventions[42].

Our present results are based on one type of movement sensor and placements to capture the essential characteristics of an

infant's spontaneous motor ability[22]. It is possible that different sensor placements[69,70] and/or modalities could capture some peripheral movements at even higher fidelity. However, we are not aware of wireless technology that would be currently available for constructing such viable solutions in clinical and home environments. The present work only focuses on the range of movement patterns during infancy, while different movement types will continue qualitative development beyond the infantile period by adapting the variability of motor control to changing developmental needs[12,67], considered to also be essential for later cognitive development[12,71,72]. Any other motor ability classification will need new training of the respective classifiers. Finally, our study shows clearly the feasibility of tracking motor ability and building growth charts from their metrics or combined indices like BIMS; However, establishing clinically accepted reference values will require larger prospective collection of normative data, preferably across different cultures to accommodate possible cultural differences[28] and possible changes over time[73].

Our work demonstrates a fully functional solution for motor ability tracking, which is already being used in the first clinical research trials. However, there are several technical, practical, and regulatory steps before it can be applied widely in the clinic. First, more experience is needed from different medical centers, health care environments, and user groups to define the practical issues related to infant recordings and analysis logistics. For instance, it will be essential to study the utility of such an approach in early diagnostics and follow-up of neurodevelopmental compromise or therapeutic efficacies. Here we examined how recording length affects the accuracy of BIMS estimate, however, future studies will be needed to fully explore the relationships between recording times and study results, which will likely vary between clinical questions and metrics of interest[74–76]. Another aspect of our method's sensitivity is to see the minimal detectable change in BIMS or any other MAIJU-derived metric, which can only be studied in appropriately designed longitudinal cohorts. It will also be essential to evaluate the cost-benefit questions that are partly specific to different health care settings. An important aspect of that work will be to establish the added clinical or scientific value of the new method relative to all the existing methods, such as AIMS. Second, coordinated manufacturing and compilation of the technological components in the full MAIJU solution is needed to support higher production volumes. Third, sufficiently large normative datasets are needed from different health care settings and diagnostic groups to test the practical diagnostic reliability[14,19,77], as well as to establish internationally approved reference data for both the motor ability metrics and the BIMS growth charts. Fourth, while clinical research can be carried out with investigational permissions, a prospective clinical use as a registered medical device needs an accurate definition of user cases[78]. This will likely involve the development of the respective legislations to accommodate continual learning that characterizes medical devices based on machine learning algorithms[79]. Since wearables of this kind are conceptually novel and they potentially change health care processes, it is also essential to define the user cases together with the relevant communities in neurodevelopmental research and medical care.

In summary, the present study shows proof of concept that the full sequence of infant motor ability development, from lying supine to walking fluently, can be quantified objectively by using a wearable method combined with a motor ability description scheme that is automatically analyzed using a deep-learning-based algorithm. These novel metrics of motor ability are transparent, intuitively interpretable, and they link strongly to infant age. Moreover, the metrics can be used further in training an algorithmic estimation of maturation of infant motor ability,

BIMS, which is robust to variations in the recording length and the child's age, and it also correlates significantly to other clinical and parental assessments of infant's performance. A solution of this kind is readily automated and widely scalable to a global extent; hence it holds significant promise for the early assessment of neurodevelopmental delays, as well as providing a functional benchmark for individualized patient care or early intervention trials.

## Data availability

An example dataset of three recordings is made publicly available at https://doi.org/10.5281/zenodo.641748630. The other original (raw) movement data can be made available upon request to the authors (S.V.). The use of this dataset in further scientific work will require a data-sharing agreement with Helsinki University Hospital. Processed data, such as motility classifier outputs, can be made available upon request.

## Code availability

Data preprocessing and analysis were performed using custom Matlab codes (version 2021a). The analysis codes are made publicly available at Zenodo30 (https://doi.org/10.5281/zenodo.6417486). The video recordings were annotated with Anvil software (version 6.0; https://www.anvil-software.org/). The motility classifier was implemented as custom code using Python (version 3.6.9) and Tensorflow (version 1.12.0). The motility classifier can be run through the BABA cloud (www.babacloud.fi) with credentials that are freely available at request (M.A.).

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

## Acknowledgements

This research received funding from Academy of Finland grants 314450, 332017, 335788 (S.V. and L.M.H.), 314573, 335872, and 343498 (M.A.), 314602 (O.R.), Juselius Foundation (S.V.), Aivosäätiö (S.V., E.I., and M.A.), and Pediatric Foundation (S.V., L.H., E.I., O.R., and M.A.). We would like to thank the research nurses for their help in conducting this study, and the children and their parents for taking part in this study.

## Author contributions

Conceptualization: M.A., L.M.H., and S.V. Methodology: M.A., E.I., O.R., L.M.H., and S.V. Investigation: M.A., A.G., P.V., T.H., and A.K. Visualization: M.A. Funding acquisition: M.A., E.I., O.R., L.M.H., and S.V. Project administration: E.I., O.R., L.M.H, and S.V. Supervision: O.R., L.M.H., and S.V. Writing—original draft: M.A. and L.M.H. Writing—review and editing: all authors.

## Competing interests

The authors declare the following competing interests: E.I. is the founder and shareholder of Planno Ltd, which consults in technical textile design and manufacturing. The remaining authors declare no competing interests.
