## [Peer Review File · Communications Medicine]

Reviewers' comments:

Reviewer #1 (Remarks to the Author):

This paper describes a new automatized characterization of motor development in human infants. Based on data from 59 infants the authors demonstrate that it is possible for a computer algorithm to determine the developmental age of an infant based on movement signals obtained from accelerometers build into a jump-suit. This approach shows great promise in determining developmental abnormalities in movement patterns of infants in their natural surroundings and may therefore be of significant clinical value.

This study is based on a fundamentally brilliant idea: Using classifiers from visual observations of infants to inform machine learning algorithms to predict the developmental stage of an infant from accelerometer signals in a simple jump-suit. I would like to congratulate the authors for this brilliant idea. The use of deep learning to automatize and simplify complex information in the clinic is blooming and well in line with the current trend of using computer algorithms as an integrated part of clinical diagnostics and decision making. In this context I like the balanced and well-thought trough approach that the authors show here.

The measurements and analysis have been well carried out as far as I can determine. The manuscript is well written and results are clearly presented. The Introduction and Discussion cover alle relevant issues and adequately relate the findings to other work in the field. However, I do have a few queries in relation to the population of infants which I would like to see clarified. I have also spotted a few minor issues in the text:

In Introduction l. 58 it says 64 children but in Methods it says 59 infants who were recorded at 64 sessions. Please, correct this in Introduction.

Please clarify the criteria for inclusion and exclusion of infants.

There are data from 59 infants. This is kind of a strange number of infants. Probably it reflects the reality of recruitment and the necessity of publishing data at some point – so no real prior calculations or decisions regarding how many infants that would be necessary for this kind of study. I think the authors need to point this out specifically and give it some thought whether this may be a potential problem for the validity of the approach.

It seems to be a convenience sample of infants from the NICU – some were premature, some had perinatal asphyxia and some appeared to had signs that had raised suspicion of brain injury at some point. This makes me wonder to what an extend this material provides information about typical development or whether there is some component of atypical/abnormal development which may influence the outcome and jeopardize the use of the technique for spotting differences between a typically developing infant and abnormal development.

How was it ensured that the cohort of infants covered a relevant age span? Was this done ad Hoc or was some kind of model/algorithm used in order to ensure that there were no gaps in the dataset which could skew the distribution in one way or the other.

I am a little worried that the five infants who were recorded twice are included with both measurements in the material. Strictly speaking those 10 measurements out of the 64

measurements are not independent. I don't think it will seriously affect the results in any way, but I believe that it would be more correct to include only one measure from those 5 infants (to have 59 independent datasets) and use the second measurements from the 5 children in a later longitudinal study. Could I ask the authors to comment on this?

The authors do present the data from the 5 infants in the end of the Result section as a kind of pilot longitudinal study. However, recording at two sessions in only 5 infants is not a very substantial material and I don't find it of much value in the present study. The same goes for the inclusion of the 4 infants who show abnormal development and who have apparently also received a clinical diagnosis. Again, this is a very limited data material and I don't think it can be used to make any real conclusions. I would suggest to leave these data out at present and put them together with new data in one or two later studies. I find the data from the 54/59 infants sufficiently strong as it is and find it a pity if the inclusion of these two small data sets is seen by some readers as a weak part in an otherwise convincing study.

Reviewer #2 (Remarks to the Author):

Wearable sensors offer a novel opportunity to measure infant movement and posture in a continuous, fairly unobtrusive way across hours or even days. There is potential for more sensitive and specific measures of developmental status as well as for more accurate identification of atypical development. This work is novel and important toward these goals, and is foundational. I commend the authors on a thorough and thoughtful approach. However, while this paper represents necessary foundational work and is important, it is a first step toward these goals, not a definitive "end product". As such, I suggest that the authors need to soften their conclusions a bit, as much more validation and development is needed before achieving an "end product". In fact, I would suggest that it is important for you to highlight the work that still needs to be done in order to encourage funding sources to support such work! I am excited for the future of your work and for the work of others in this area (who will reference this work). My specific comments follow below. My comments are provided with the intention of engaging in discussion about your important and interesting work. Respectfully, Beth A. Smith, PT, PhD

1. General comment. Motility is commonly used in reference to intestinal tract movement and or movement of sperm, while mobility is commonly used in reference to humans moving through the environment. It struck me as odd to read motility here, I would have used mobility. That said, I am not insisting you change it, I respect that this is your work and it is your decision how you want to name it/describe it.

2. I have one major concern: data when infants were picked up and carried were excluded from analyses for creation of the classifier algorithms. Given the authors are advocating in the Introduction for "This could be solved with an objective measurement of spontaneous behavior at home, the most ecologically valid environment", I believe it is necessary to include discussion of the following two points: 1. How will data when infants are picked up/carried be identified and removed from in-home recordings? 2. If they are not removed, accuracy will be lower than what is presented here. Please see the following relevant references: Worobey, J. Physical activity in infancy: developmental aspects, measurement, and importance. *The American Journal of Clinical Nutrition* 99, 729S–33S (2014) and Zhou, J., Schaefer, S. Y. & Smith, B. A. Quantifying Caregiver Movement when Measuring Infant Movement across a Full Day: A Case Report. 19, 2866 (2019). This seems to

be addressed in the section “Development of the carrying detection classifier” but I am confused as the data in this section appear to be analyzed separately. It seems that ~50% of picked up/moved around data can be filtered out automatically. But that will leave ~50%, as opposed to removing all of it as was done to train classifiers. I also did not see inter-rater reliability reported for video coding of the data presented in Fig. S5: Decision tree diagram of the active carrying detection (ACD) annotations.

3. Abstract: “and it also provides a key functional benchmark for individualized patient care, including any early intervention trials.” Please state that it provides a foundation for creation of a functional benchmark for individualized patient care and early intervention trials, see my further comments of this nature below.

Introduction:

4. “Only a small minority of these infants will be eventually diagnosed with severe disabilities, such as cerebral palsy, while a larger portion of infants will develop mild or moderate neurocognitive impairments, such as disorders of communication and attention.” Point 1: Cerebral palsy can range from mild to severe disability, it is not only a severe disability, and this sentence needs a citation. Further, percentage values would be more informative for the reader than use of “small minority” and “larger portion”. Point 2: I don’t think it is clear in this paragraph that although early intervention is recommended for these infants it is difficult to identify who needs it because most of the diagnoses are made well after early intervention should have occurred. While diagnosis of cerebral palsy is now possible much earlier than it used to be (now ~3-4 months post-term due to the General Movements Assessment and CP Guidelines as described in the Hadders-Algra review, citation 3), most other neurodevelopmental diagnoses are not made until around 2 years of age. Point 3: Whether infants who are diagnosed later with disorders of communication and attention would show measurable motor development differences early in life is questionable. While I appreciate that motor, cognitive, and language development are intertwined and interrelated, any differences in motor development in infants later diagnosed with disorders of communication and attention may be very subtle and impossible to detect among the large variability of typical development. Again, this all speaks to the necessary future work required. I see the need for more sensitive and specific identification of early motor delay/impairment but there is much work required before sensor data can be used to predict neurodevelopmental outcomes and target early intervention.

5. “The early development of an infant’s motor abilities provides an essential framework in the developmental cascade leading into language and cognitive abilities^{4–9}”. Many of the cited studies are based on associations, not cause and effect. While it makes sense that early motor ability supports language and cognitive development, it also makes sense that early language and cognitive development support motor development. Please consider changing “leading into” into “related to”.

6. “However, milestone assessment does not quantitate the spontaneous motility of infants, and it also ignores the wide variability that characterizes natural motor development^{13,14}.” Milestone assessment has very large ranges for onset of milestones, which makes it more difficult to identify atypical development early and provide early intervention. But I wouldn’t say it “ignores” the variability, I would say the variability limits its usefulness. And while it is true that milestone assessment does not “quantitate the spontaneous motility of infants” it is not clear why this missing information is necessary.

7. “Here, we set out to construct and validate a generalizable, scalable, and effective method to measure infants’ spontaneous motility across all milestone levels of infant motor development from lying supine to fluent independent walking.” I appreciate that a generalizable, scalable, and effective method is the goal of this work, however I do not see where these three items were specifically evaluated in this paper. For example, how did you show with these results that it is generalizable, and what criteria would have led you to determine that it was not generalizable? Please change to “Our overall goal is to...” or something similar.

Methods

8. Re: cross-sectional vs longitudinal data. Initially it was unclear to me if these were cross-sectional or longitudinal data, for example at the end of the Introduction “across all milestone levels of infant motor development from lying supine to fluent independent walking” had me expecting longitudinal data. Without longitudinal data sensitivity to change is not known for individual trajectories, nor is variability within individual trajectories. For example, I would expect the elementary walking of one infant to be more similar to the later fluid walking of the same infant, and thus more difficult to distinguish from one another than elementary walking of one infant and fluid walking of another infant. And at what point in the transition from elementary to fluid walking would the transition be identified? I suspect it might be unstable between the two categories for some period of time before transitioning within one infant. Please make it more clear that these are primarily cross-sectional data that categories are being determined from, for these reasons.

9. Re: cross-sectional vs longitudinal data. At the end of the paper it says 5 infants had repeated measures...how were repeated measures accounted for in correlation measures?

10. Were different types of crawling/scotting captured? Or only hands and knees?

11. Just before Discussion: Please add for future work that the approach will need to be validated for different at risk or neurodevelopmental diagnoses. Four infants with delay represents a good proof of concept but is not sufficient.

12. The AIMS has norms only for birth- 18 months and is not advised to be used past 18 months of age. Please mention this when you discuss “ceiling effects” of motor development in your sample.

13. I don’t think the question of how much data you need to capture an infant’s full repertoire of movement skills has been answered yet here (Figure 4C and associated text). As you discuss, rare activities require longer recording periods to capture than frequent activities. Further, recording period may vary by age, and may require specified caregiver prompts for certain activities at certain ages (e.g., placing a toy to encourage crawling). Again, these are things that will need to be answered with future work. Please see these relevant citations: Abrishami, M. S., Mert, M.... Identification of Developmental Delay in Infants Using Wearable Sensors: Full-Day Leg Movement Statistical Feature Analysis. *IEEE journal of translational engineering in health and medicine* 7, 2800207–7 (2019), Franchak, J. M., Scott, V. & Luo, C. A Contactless Method for Measuring Full-Day, Naturalistic Motor Behavior Using Wearable Inertial Sensors. *Front Psychol* 12, 701343 (2021) and Franchak, J. M. Changing Opportunities for Learning in Everyday Life: Infant Body Position Over the First Year. *Infancy* 1–23 (2018) doi:10.1111/infa.12272.

14. For the parent survey mentioned just before discussion, please describe the survey that was used. Was it custom made? Was it retrospective or prospective? What periods of time were analyzed, and how were questions asked? Different biases and errors are expected with these different methods.

15. Please also see these relevant citations about estimating infant/toddler position using sensors: Jeong, H. et al. Miniaturized wireless, skin-integrated sensor networks for quantifying full-body movement behaviors and vital signs in infants. *Proc National Acad Sci* 118, e2104925118 (2021) and Kwon, S., Zavos, P., Nিকেle, K., Sugianto, A. & Albert, M. V. Hip and Wrist-Worn Accelerometer Data Analysis for Toddler Activities. *International Journal of Environmental Research and Public Health* 16, 2598–12 (2019).

DISCUSSION

16. “We constructed and validated a clinically applicable method for accurate, objective and quantitative tracking of an infant’s gross motor performance throughout the full developmental sequence from lying supine to fluent walking.” --- In my opinion, “clinically applicable” has not yet been shown. Clinically applicable would mean that it could be used within a typical clinical context: the suit provided to parents by clinicians (not researchers with funding and time and training) with the resulting information interpreted and used by clinicians in their clinical assessment. I agree you can talk about the points of your system that support its potential for clinical application, but there is a cost/benefit question that is fundamental to clinical application that has not been answered yet.

17. “This predictor yields a novel, transparent summary measure of motor development, BIMS, which holds promise as an objective universal metric in early neurodevelopmental assessment, supporting individualized neurodevelopmental care as well as benchmarking of clinical trials^{31–33}”. This sentence is referring to predicting age, which is not something that needs to be “predicted” in neurodevelopmental care or clinical trials. It is a good first step that BIMS correlates with both AIMS and age, and more strongly with AIMS than age. Further, while it is promising that your analysis captures that time spent in prone decreases as infants get older, this is not the level of analysis that will be required to achieve the goals you refer to. In order to support individualized neurodevelopmental care and benchmarking of clinical trials it will need to provide more useful data than AIMS or other easy to administer, short, inexpensive assessments. It will need to be more sensitive and specific at classifying development and/or more accurate at identifying atypical development than existing tools.

18. “However, most of the prior work has been constrained by practical limitations in the recording configuration and data analytics, such as using only one or two accelerometer sensors to quantify only the gross amounts of movements^{20,34,38}. This approach ignores the actual hallmark of motor development, the wide qualitative spectrum of different motility types in the child’s spontaneous behavior^{13,14,25}”. To me “ignores” sounds intentional and condescending, when these cited papers are first steps in developing methods to measure aspects of motor development or identify atypical development early in life. I agree with highlighting the unique aspects and strengths of your paper, but I think you can do it without claiming other researchers are “ignoring” something just because that was not the focus of a given paper. It is often discussed as something that is important and needs to be addressed in future work. See relevant paper: Smith, B. A., Trujillo-Priego, I. A., Lane, C. J., Finley, J. M. & Horak, F. B. Daily Quantity of Infant Leg Movement: Wearable Sensor Algorithm and Relationship to Walking Onset. *Sensors*, 15, 19006–19020 (2015).

19. "For instance, BIMS provides a straightforward measure for the challenging but topical studies on the developmental cross-domain interactions, such as is seen between infant neuromotor development, later cognitive outcomes and/or environmental enrichment interventions³³." Please add that sensitivity to change, "minimum detectable change", of BIMS needs to be determined from longitudinal data before it can be used as an outcome measure.

Reviewer #5 (Remarks to the Author):

The current manuscript demonstrates a new approach to studying the development of human motility using a novel methodology that includes the innovation of wearable technology and deep-learning algorithms to detect children's posture and movement. The authors suggest that their new technique is "valuable for early diagnosis and tracking of neurodevelopmental delays" and "provides a key functional benchmark for individualized patient care."

I am a big supporter of using novel technologies to track development and to gather the information that will allow the description of development in multiple dimensions rather than a narrow milestone chart. I also think the authors' approach to this challenge is correct, their wearable is a non-expensive technology and their deep-learning algorithms are interesting. The combination of both certainly has the potential (but not more than that at the moment) to revolutionise tracking and diagnosis of neurodevelopmental disorders\delays. Using this rich data can provide new perspectives that even human experts are lacking. Finally, the authors did a very good job in describing and plotting their data and did most of the required statistical tests. It's definitely shown that the authors worked thoroughly and are very transparent about all their data. This should not be taken for granted. As an advocate of open science, I thank the authors for this and encourage them to do so in the future.

That said, the manuscript needs a lot of work. It's not clear what this paper is about, no clear questions or hypotheses regarding human development, and the introduction and discussion are a total mess. All these issues make it hard to evaluate the current contribution (besides the methodology, see below for more explanation) of this work to the research of human development, neurodevelopmental disorders, or medicine. Thus, the current form of the paper is unpublishable (and even hard to review), despite its importance.

Major Issues:

a. The biggest issue for me with this paper is that I do not understand what it is about and therefore, I'm having a hard time evaluating its contribution.

Is it about the novel technology? If so – the authors should publish a method paper in which they provide all the details about the technology and only show a comparison between their tool to the manual annotations. All other discussions, data, figures, etc. are not relevant and should be dropped. The authors do not really show this is effective for the early diagnosis of developmental disorders. It's not even clear why it should be at the moment, and what developmental disorders? and data from 4 atypical developing children does not prove it. Currently, the authors just bombarded the reader with a lot of details that do not even make sense. For example, why did the authors' selected movements that have so low inter-raters agreement? What can we say about these movements if even two humans cannot agree on what they are? Why are these checked in these fancy algorithms? What is the motivation for all of it? For now, it just seems like the authors tried their (really cool) algorithm on several postures and movements that work but no sense in

terms of developmental cascades, neurodevelopmental disorders, etc.

b. I agree with the authors that it is very powerful to get movement data in a free environment, where infants are not constrained to the lab. Moreover, rich data from daily activities can really contribute to our understanding of infants' motor development. But the authors do none of these. They do not describe the environment in which these movements occurred, they do not motivate any of the movements\postures they selected (just to clarify - they do motivate but not in terms of free, real-world environment, why are that specific movements\posture will be interesting to test with the wearables in the real-world environment rather than with videos in the lab?). Same for the neurodevelopmental delays – it is true that potentially these wearables can be used to detect delays. But how? The authors did not really test it, and did not associate the movements\postures to specific developmental delays (Again, testing few children with delays is not publishable, they should make a real experimental study).

c. The authors did not convince me that the data they get from the wearables and their algorithms provide new information on human development that we do not know from previous work. The authors only convinced me that their technology can capture some postures that humans can rate by looking at a video. This is not a finding that is worth publication in Communications Medicine. It is an outstanding computer-science paper, but this is not a breakthrough in human development. In my opinion, the main reason for that is the use of supervised learning in this paper. By using supervised learning, the authors train the algorithms to capture things they wanted to look at. This is certainly nice and the accuracies are very impressive, but what is the novelty here? Besides the technology, what is the contribution of this paper?

d. The authors' BIMS score does not provide any insights about developmental delays and the authors are not convincing in explaining the rationale behind it and why it is needed. Similar to the other work in this paper – the authors simply show that their measures correlate with existing measures. This is good and encouraging but what is the big advantage of using BIMS? Why is it better than existing methods?

e. The authors argue that “our work demonstrates a fully functional solution that is already used in a clinical research environment” – this paper does not provide evidence for that. The authors need a real clinical-trial experiment to back up all their arguments in the discussion section.

There are several minor issues and many missing references and discussion or previous empirical evidence. However - the authors must fix the above major issues before those can be given. The paper is not focused, there are too many ideas, too many issues and none of them (Besides the technology itself) is not at a level of publication.

February 23, 2022

Rebuttal letter

Re: "*Intelligent wearable allows out-of-the-lab tracking of infant motility development*"

Reviewer #1 (Remarks to the Author):

This paper describes a new automatized characterization of motor development in human infants. Based on data from 59 infants the authors demonstrate that it is possible for a computer algorithm to determine the developmental age of an infant based on movement signals obtained from accelerometers build into a jump-suit. This approach shows great promise in determining developmental abnormalities in movement patterns of infants in their natural surroundings and may therefore be of significant clinical value.

This study is based on a fundamentally brilliant idea: Using classifiers from visual observations of infants to inform machine learning algorithms to predict the developmental stage of an infant from accelerometer signals in a simple jump-suit. I would like to congratulate the authors for this brilliant idea. The use of deep learning to automatize and simplify complex information in the clinic is blooming and well in line with the current trend of using computer algorithms as an integrated part of clinical diagnostics and decision making. In this context I like the balanced and well-thought trough approach that the authors show here.

The measurements and analysis have been well carried out as far as I can determine. The manuscript is well written and results are clearly presented. The Introduction and Discussion cover alle relevant issues and adequately relate the findings to other work in the field. However, I do have a few queries in relation to the population of infants which I would like to see clarified. I have also spotted a few minor issues in the text:

Q1.1. In Introduction l. 58 it says 64 children but in Methods it says 59 infants who were recorded at 64 sessions. Please, correct this in Introduction.

A1.1. This has been corrected in the revised manuscript.

Q1.2. Please clarify the criteria for inclusion and exclusion of infants.

A1.2. These have been clarified in Methods, section Participants and recordings.

There are data from 59 infants. This is kind of a strange number of infants. Probably it reflects the reality of recruitment and the necessity of publishing data at some point – so no real prior calculations or decisions regarding how many infants that would be necessary for this kind of study. I think the authors need to point this out specifically and give it some thought whether this may be a potential problem for the validity of the approach.

It seems to be a convenience sample of infants from the NICU – some were premature, some had perinatal asphyxia and some appeared to had signs that had raised suspicion of brain injury at some point. This makes me wonder to what an extend this material provides information about typical development or whether there is some component of atypical/abnormal development which may influence the outcome and jeopardize the use of the technique for spotting differences between a typically developing infant and abnormal development.

Q1.3. How was it ensured that the cohort of infants covered a relevant age span? Was this done ad Hoc or was some kind of model/algorithm used in order to ensure that there were no gaps in the dataset which could skew the distribution in one way or the other.

A1.3. We think you are absolutely right! The aim of this study was not to establish normative values for clinical purposes and we have now clarified this in the manuscript. This work aims to cover a detailed description on the development process for a holistic methodological solution (MAIJU) for tracking infants' spontaneous motor behavior.

The present cohort represents infants who have primarily participated in other ongoing clinical trials (e.g. mild perinatal asphyxia, preterm infants) and, in addition, volunteered to participate in this study. Accordingly, the infants do represent a likely target group of the MAIJU solution.

Based on the general knowledge and clinical experience, we defined the relevant age span so that the youngest infants would be unable to move around, whereas the oldest infants would have reached a fluent skill in walking. This then served as the framework in our motor ability description scheme. It is plausible that distributions (e.g., Fig 4A) will shift with accumulating data and alter according to the different patient cohorts. Indeed, such prospective collections of normative/reference datasets are already running, and their need is highlighted in the Discussion:

“Third, sufficiently large normative datasets are needed from different health care settings and diagnostic groups to test the practical diagnostic reliability 11,16,64, as well as to establish internationally approved reference data”

Q1.4. I am a little worried that the five infants who were recorded twice are included with both measurements in the material. Strictly speaking those 10 measurements out of the 64 measurements are not independent. I don't think it will seriously affect the results in any way, but I believe that it would be more correct to include only one measure from those 5 infants (to have 59 independent datasets) and use the second measurements from the 5 children in a later longitudinal study. Could I ask the authors to comment on this?

A1.4. The focus of the experiment in Fig 4B is to study the performance of the algorithm in being able to predict an infant's age based on the MAIJU recording. I.e., it is not a measure that reflects infant development, but the algorithm itself. For such purpose, the data samples are not required to be independent. In contrast, the experiment depicted in Fig 4D is correlated against developmental scores, and requires accounting for non-independent measures.

Q1.5. The authors do present the data from the 5 infants in the end of the Result section as a kind of pilot longitudinal study. However, recording at two sessions in only 5 infants is not a very substantial material and I don't find it of much value in the present study. The same goes for the inclusion of the 4 infants who show abnormal development and who have apparently also received a clinical diagnosis. Again, this is a very limited data material and I don't think it can be used to make any real conclusions. I would suggest to leave these data out at present and put them together with new data in one or two later studies. I find the data from the 54/59 infants sufficiently strong as it is and find it a pity if the inclusion of these two small data sets is seen by some readers as a weak part in an otherwise convincing study.

A1.5. Thank you for the thoughtful comment. We agree that the presented data regarding longitudinal and abnormal data points is not substantial enough to make real conclusions based on them. Furthermore, their visual emphasis might dissuade readers from understanding the reason for their inclusion (which in our original view was to serve as “sanity checks”), and make false conclusions about the intent of our paper. Based on these reasons, we have omitted the outlier data points in Fig 4B, and removed the visualization of longitudinal connections. As the use of the longitudinal data points in itself is not a problem in this case (see A1.4), they are still included in the Fig 4B.

Reviewer #2 (Remarks to the Author):

Wearable sensors offer a novel opportunity to measure infant movement and posture in a continuous, fairly unobtrusive way across hours or even days. There is potential for more sensitive and specific measures of developmental status as well as for more accurate identification of atypical development. This work is novel and important toward these goals, and is foundational. I commend the authors on a thorough and thoughtful approach. However, while this paper represents necessary foundational work and is important, it is a first step toward these goals, not a definitive “end product”. As such, I suggest that the authors need to soften their conclusions a bit, as much more validation and development is needed before achieving an “end product”. In fact, I would suggest that it is important for you to highlight the work that still needs to be done in order to encourage funding sources to support such work! I am excited for the future of your work and for the work of others in this area (who will reference this work). My specific comments follow below. My comments are provided with the intention of engaging in discussion about your important and interesting work. Respectfully, Beth A. Smith, PT, PhD

Q2.1. General comment. Motility is commonly used in reference to intestinal tract movement and or movement of sperm, while mobility is commonly used in reference to humans moving through the environment. It struck me as odd to read motility here, I would have used mobility. That said, I am not insisting you change it, I respect that this is your work and it is your decision how you want to name it/describe it.

A2.1. Thank you for the comment. We appreciate the terminological challenge when approaching an old target using a conceptually novel approach. Motivated by this comment, we have now re-performed a quick international survey among native experts in the field, and we decided to use the term “motor ability” instead of motility.

According to our native advice, the term “mobility” would not cover posture, which is an essential component of assessment done with MAIJU.

Q2.2. I have one major concern: data when infants were picked up and carried were excluded from analyses for creation of the classifier algorithms. Given the authors are advocating in the Introduction for “This could be solved with an objective measurement of spontaneous behavior at home, the most ecologically valid environment”, I believe it is necessary to include discussion of the following two points:

1. How will data when infants are picked up/carried be identified and removed from in-home recordings? 2. If they are not removed, accuracy will be lower than what is presented here. Please see the following relevant references: Worobey, J. Physical activity in infancy: developmental aspects, measurement, and importance. *The American Journal of Clinical Nutrition* 99, 729S–33S (2014) and Zhou, J., Schaefer, S. Y. & Smith, B. A. Quantifying Caregiver Movement when Measuring Infant Movement across a Full Day: A Case Report. 19, 2866 (2019). This seems to be addressed in the section “Development of the carrying detection classifier” but I am confused as the data in this section appear to be analyzed separately. It seems that ~50% of picked up/moved around data can be filtered out automatically. But that will leave ~50%, as opposed to removing all of it as was done to train classifiers. I also did not see inter-rater reliability reported for video coding of the data presented in Fig. S5: Decision tree diagram of the active carrying detection (ACD) annotations.

A2.2. We understand that this part of the study was unclear. As a part of our protocol, the in-home recordings are instructed to contain a designated “play time” (~1 hour), during which the parents are encouraged to let the infants play independently as much as possible. Parental care and contact is naturally allowed during this time period, and the active carrying detector (ACD) is utilized to filter out time periods of movement that are a result of this. In all of the subsequent analyses, the ACD is applied (see Fig S8) to improve the results compared to analyzing mixtures of play and incidental carrying during a playtime. We have now described this more clearly in the ACD section within Methods.

Regarding the ACD annotations, unfortunately, we don't have multiple annotations per infant and hence cannot provide the inter-rater agreement estimates. However, this annotation task was perceived to be easy (i.e., more on a par with the Posture annotations than with the Movement annotations). We agree that future work will be useful in establishing and perfecting the ACD in different user scenarios, but that would go far beyond the scope of the present study.

Q2.3. Abstract: “and it also provides a key functional benchmark for individualized patient care, including any early intervention trials.” Please state that it provides a foundation for creation of a functional benchmark for individualized patient care and early intervention trials, see my further comments of this nature below.

A2.3. Thank you. The proposed phrasing is now added, it is certainly more on-point with what we want to communicate.

Introduction:

Q2.4. “Only a small minority of these infants will be eventually diagnosed with severe disabilities, such as cerebral palsy, while a larger portion of infants will develop mild or moderate neurocognitive impairments, such as disorders of communication and attention.” Point 1: Cerebral palsy can range from mild to severe disability, it is not only a severe disability, and this sentence needs a citation. Further, percentage values would be more informative for the reader than use of “small minority” and “larger portion”. Point 2: I don't think it is clear in this paragraph that although early intervention is recommended for these infants it is difficult to identify who needs it because most of the diagnoses are made well after early intervention should have occurred. While diagnosis of cerebral palsy is now possible much earlier than it used to be (now ~3-4 months post-term due to the General Movements Assessment and CP Guidelines as described in the Hadders-Algra review, citation 3), most other neurodevelopmental diagnoses are not made until around 2 years of age. Point 3: Whether infants who are diagnosed later with disorders of communication and attention would show measurable motor development differences early in life is questionable. While I appreciate that motor, cognitive, and language development are intertwined and interrelated, any differences in motor development in infants later diagnosed with disorders of communication and attention may be very subtle and impossible to detect among the large variability of typical development. Again, this all speaks to the necessary future work required. I see the need for more sensitive and specific identification of early motor delay/impairment but there is much work required before sensor data can be used to predict neurodevelopmental outcomes and target early intervention.

A2.4. Point 1: Thank you for this comment. We agree that the CP phenotype is highly variable. In order to point out this important fact we have modified the text as “..infants will be eventually diagnosed with major disabilities, such as the severe types of cerebral palsy, while a larger portion of infants...”

We have also added here a reference on cerebral palsy ([3] Bax et al. 2005).

We have also provided two other references which represent examples of clinical risk groups and the wide range of outcomes from typical development to a spectrum of neurodevelopmental problems. ([4] Rancken et al. 2021, [5] Serenius et al. 2016).

We agree that percentage values would be more informative for the reader than use of “small minority” and “larger portion”. However, there is no international consensus on what criteria should be used to define e.g. the wide spectrum of neurodevelopmental impairment, and accordingly, any percentage approximation would be highly dependent on the source of information.

Point 2: In the clinical practice identifying the infants with a high risk of CP during the first months of life is based on a combination of detailed patient history, validated neurological examination or neuromotor assessment, and brain imaging. Based on available research evidence, the best three tools to detect high risk of CP before the corrected age of five months old are neonatal magnetic resonance

imaging (MRI), the Prechtl Qualitative Assessment of General Movements (GMs), and the Hammersmith Infant Neurological Examination (HINE). After the corrected age of five months old, the recommended tools are brain MRI, the HINE, and standardized motor assessment tools. Early identification of high risk of CP promotes early exploration of available treatment options and early intervention that aim to enhance innate brain plasticity for improved functional outcome. Accordingly, early intervention should be launched if a high risk of CP is suspected¹. We fully agree that targeting early intervention in a wide range of neurodevelopmental disorders is a very complex issue. We would prefer not to go into all these aspects since the focus of this manuscript was not to establish normative values for clinical purposes but to provide a detailed description on the development process for a holistic methodological solution for tracking infant's spontaneous motor behavior.

¹Novak I, Morgan C, Adde L, et al. Early, Accurate Diagnosis and Early Intervention in Cerebral Palsy: Advances in Diagnosis and Treatment. *JAMA Pediatr.* 2017;171(9):897–907.

Point 3: We fully agree that extensive more future work is required.

Q2.5. “The early development of an infant’s motor abilities provides an essential framework in the developmental cascade leading into language and cognitive abilities^{4–9}”. Many of the cited studies are based on associations, not cause and effect. While it makes sense that early motor ability supports language and cognitive development, it also makes sense that early language and cognitive development support motor development. Please consider changing “leading into” into “related to”.

A2.5. Thank you. The phrasing was changed as suggested.

Q2.6. “However, milestone assessment does not quantitate the spontaneous motility of infants, and it also ignores the wide variability that characterizes natural motor development^{13,14}.” Milestone assessment has very large ranges for onset of milestones, which makes it more difficult to identify atypical development early and provide early intervention. But I wouldn’t say it “ignores” the variability, I would say the variability limits its usefulness. And while it is true that milestone assessment does not “quantitate the spontaneous motility of infants” it is not clear why this missing information is necessary.

A2.6. Thank you. We have now modified the phrasing:

“However, milestone assessment does not quantitate the spontaneous motor ability of infants, and it is not sensitive to the wide variability that characterizes natural motor development”

Q2.7. “Here, we set out to construct and validate a generalizable, scalable, and effective method to measure infants’ spontaneous motility across all milestone levels of infant motor development from lying supine to fluent independent walking.” I appreciate that a generalizable, scalable, and effective method is the goal of this work, however I do not see where these three items were specifically evaluated in this paper. For example, how did you show with these results that it is generalizable, and what criteria would have led you to determine that it was not generalizable? Please change to “Our overall goal is to...” or something similar.

A2.7. Thanks, we agree. This has been rephrased

“Here, we set an overall goal to construct and validate a....”

Methods

Q2.8. Re: cross-sectional vs longitudinal data. Initially it was unclear to me if these were cross-sectional or longitudinal data, for example at the end of the Introduction “across all milestone levels of infant motor development from lying supine to fluent independent walking” had me expecting

longitudinal data. Without longitudinal data sensitivity to change is not known for individual trajectories, nor is variability within individual trajectories. For example, I would expect the elementary walking of one infant to be more similar to the later fluid walking of the same infant, and thus more difficult to distinguish from one another than elementary walking of one infant and fluid walking of another infant. And at what point in the transition from elementary to fluid walking would the transition be identified? I suspect it might be unstable between the two categories for some period of time before transitioning within one infant. Please make it more clear that these are primarily cross-sectional data that categories are being determined from, for these reasons.

A2.8. We have now highlighted the primarily cross-sectional nature of the study cohort in the Methods. In the section Study design:

“A primarily cross-sectional cohort of infants “

and in the section Participants and recordings:

“While this cohort was primarily cross-sectional, five infants...”

Q2.9. Re: cross-sectional vs longitudinal data. At the end of the paper it says 5 infants had repeated measures...how were repeated measures accounted for in correlation measures?

A2.9. Please see our answers to the first reviewer A1.4 and A1.5:

A1.4. The focus of the experiment in Fig 4B is to study the performance of the algorithm in being able to predict an infant’s age based on the MAIJU recording. I.e., it is not a measure that reflects infant development, but the algorithm itself. For such purpose, the data samples are not required to be independent. In contrast, the experiment depicted in Fig 4D is correlated against developmental scores, and requires accounting for non-independent measures.

A1.5. Thank you for the thoughtful comment. We agree that the presented data regarding longitudinal and abnormal data points is not substantial enough to make real conclusions based on them. Furthermore, their visual emphasis might dissuade readers from understanding the reason for their inclusion (which in our original view was to serve as “sanity checks”), and make false conclusions about the intent of our paper. Based on these reasons, we have omitted the outlier data points in Fig 4B, and removed the visualization of longitudinal connections. As the use of the longitudinal data points in itself is not a problem in this case (see A1.4), they are still included in the Fig 4B.

Q2.10. Were different types of crawling/scooting captured? Or only hands and knees?

A2.10. We observed natural, spontaneous movements in infants, and our motor ability description scheme was designed to allow annotating every single time frame into some of those categories. One of the major tasks in our study was to develop such a description scheme that allows continuous annotation in a sufficiently unambiguous manner. Please see the Fig2B for graphic illustrations of the motor performances, and Table S1 for their verbal description.

Within our annotated dataset, most typical crawling pattern was certainly on hands and knees, but small amounts of movement that could be considered “bear walking” or “sideways scooting” was also present in the material, and both would be considered as “crawl posture” + “elementary”/“fluent” (i.e., the same as typical crawling). We can certainly see the potential for a sub-classifier that tries to differentiate the detected “crawl posture” + “fluent” movement into more specific crawling/scooting modes, but with our current dataset this is not yet plausible.

Q2.11. Just before Discussion: Please add for future work that the approach will need to be validated for different at risk or neurodevelopmental diagnoses. Four infants with delay represents a good proof of concept but is not sufficient.

A2.11. This has been added to the Discussion.

“For instance, it will be essential to study the utility of such an approach in early diagnostics and follow-up of neurodevelopmental compromise or therapeutic efficacies. “

Q2.12. The AIMS has norms only for birth- 18 months and is not advised to be used past 18 months of age. Please mention this when you discuss “ceiling effects” of motor development in your sample.

A2.12. Good point! We have now added this mention to the manuscript in the relevant section “Development of the BABA Infant Motor Score (BIMS) metric”

Q2.13. I don’t think the question of how much data you need to capture an infants full repertoire of movement skills has been answered yet here (Figure 4C and associated text). As you discuss, rare activities require longer recording periods to capture than frequent activities. Further, recording period may vary by age, and may require specified caregiver prompts for certain activities at certain ages (e.g., placing a toy to encourage crawling). Again, these are things that will need to be answered with future work. Please see these relevant citations: Abrishami, M. S., Mert, M.... Identification of Developmental Delay in Infants Using Wearable Sensors: Full-Day Leg Movement Statistical Feature Analysis. IEEE journal of translational engineering in health and medicine 7, 2800207–7 (2019), Franchak, J. M., Scott, V. & Luo, C. A Contactless Method for Measuring Full-Day, Naturalistic Motor Behavior Using Wearable Inertial Sensors. Front Psychol 12, 701343 (2021) and Franchak, J. M.

Changing Opportunities for Learning in Everyday Life: Infant Body Position Over the First Year. Infancy 1–23 (2018) doi:10.1111/infa.12272.

A2.13. Thank you, very important point! We have now added to the Discussion this sentence (with the above mentioned references):

“Here we examined how recording length affects the accuracy of BIMS estimate, however future studies will be needed to fully explore the relationships between recording times and study results, which will likely vary between clinical questions and metrics of interest (64-66).”

Q2.14. For the parent survey mentioned just before discussion, please describe the survey that was used. Was it custom made? Was it retrospective or prospective? What periods of time were analyzed, and how were questions asked? Different biases and errors are expected with these different methods.

A2.14. The following clarifying statement regarding the questionnaire has been added to the manucipt section “Comparison of the algorithmic output to clinical development”:

“We used a larger custom-made questionnaire to assess many aspects of the project, including MAIJU design, infant's development and parent's perception of various things. This questionnaire was delivered on paper and it was requested to be filled by the parents/caregivers by the time of MAIJU recording. For the present study, we chose two questions to be compared with MAIJU outputs: estimate on the average amount of infant’s free playing time spent in 1) crawl posture and 2) sitting posture.”

Q2.15. Please also see these relevant citations about estimating infant/toddler position using sensors: Jeong, H. et al. Miniaturized wireless, skin-integrated sensor networks for quantifying full-body movement behaviors and vital signs in infants. Proc National Acad Sci 118, e2104925118 (2021) and Kwon, S., Zavos, P., Nickele, K., Sugianto, A. & Albert, M. V. Hip and Wrist-Worn Accelerometer Data Analysis for Toddler Activities. International Journal of Environmental Research and Public Health 16, 2598–12 (2019).

A2.15. Thank you. These citations have been added to Discussion.

DISCUSSION

Q2.16. “We constructed and validated a clinically applicable method for accurate, objective and quantitative tracking of an infant’s gross motor performance throughout the full developmental sequence from lying supine to fluent walking.” --- In my opinion, “clinically applicable” has not yet been shown. Clinically applicable would mean that it could be used within a typical clinical context: the suit provided to parents by clinicians (not researchers with funding and time and training) with the resulting information interpreted and used by clinicians in their clinical assessment. I agree you can talk about the points of your system that support its potential for clinical application, but there is a cost/benefit question that is fundamental to clinical application that has not been answered yet.

A2.16. Good point. We have changed the phrasing to “potentially clinically applicable”. Added a sentence to Discussion:

“It will also be essential to evaluate the cost-benefit questions that are partly specific to different health care settings.”

Q2.17. “This predictor yields a novel, transparent summary measure of motor development, BIMS, which holds promise as an objective universal metric in early neurodevelopmental assessment, supporting individualized neurodevelopmental care as well as benchmarking of clinical trials^{31–33}”. This sentence is referring to predicting age, which is not something that needs to be “predicted” in neurodevelopmental care or clinical trials. It is a good first step that BIMS correlates with both AIMS and age, and more strongly with AIMS than age. Further, while it is promising that your analysis captures that time spent in prone decreases as infants get older, this is not the level of analysis that will be required to achieve the goals you refer to. In order to support individualized neurodevelopmental care and benchmarking of clinical trials it will need to provide more useful data than AIMS or other easy to administer, short, inexpensive assessments. It will need to be more sensitive and specific at classifying development and/or more accurate at identifying atypical development than existing tools.

Q2.17. We absolutely agree, a new method must bring benefits that justify its adoption in clinical or scientific use. We have now added this sentence to the Discussion:

“An important aspect of that work will be to establish the added clinical or scientific value of a new method relative to all the existing methods, such as the AIMS.”

Q2.18. “However, most of the prior work has been constrained by practical limitations in the recording configuration and data analytics, such as using only one or two accelerometer sensors to quantify only the gross amounts of movements^{20,34,38}. This approach ignores the actual hallmark of motor development, the wide qualitative spectrum of different motility types in the child’s spontaneous behavior^{13,14,25}”. To me “ignores” sounds intentional and condescending, when these cited papers are first steps in developing methods to measure aspects of motor development or identify atypical development early in life. I agree with highlighting the unique aspects and strengths of your paper, but I think you can do it without claiming other researchers are “ignoring” something just because that was not the focus of a given paper. It is often discussed as something that is important and needs to be addressed in future work. See relevant paper: Smith, B. A., Trujillo-Priego, I. A.,

Lane, C. J., Finley, J. M. & Horak, F. B. Daily Quantity of Infant Leg Movement: Wearable Sensor Algorithm and Relationship to Walking Onset. *Sensors*, 15, 19006–19020 (2015).

A2.18. We are sorry about this unintentionally strong wording, and we fully agree that the point may be made with an easier tone. In the revised version, we have rephrased this sentence as follows:

“Here we show that a combination of modern deep-learning signal analysis methods and a multisensory recording allow even more sensitive recognition of “

Q2.19. “For instance, BIMS provides a straightforward measure for the challenging but topical studies on the developmental cross-domain interactions, such as is seen between infant neuromotor

development, later cognitive outcomes and/or environmental enrichment interventions³³.” Please add that sensitivity to change, “minimum detectable change”, of BIMS needs to be determined from longitudinal data before it can be used as an outcome measure.

A2.19. Thank you, an excellent point. This has been added to the Discussion:

“Another aspect of our method’s sensitivity is to see the minimal detectable change in BIMS or any other MAIJU-derived metric, which can only be studied in appropriately designed longitudinal cohorts. “

Reviewer #3 (Remarks to the Author):

The current manuscript demonstrates a new approach to studying the development of human motility using a novel methodology that includes the innovation of wearable technology and deep-learning algorithms to detect children’s posture and movement. The authors suggest that their new technique is “valuable for early diagnosis and tracking of neurodevelopmental delays” and “provides a key functional benchmark for individualized patient care.”

I am a big supporter of using novel technologies to track development and to gather the information that will allow the description of development in multiple dimensions rather than a narrow milestone chart. I also think the authors’ approach to this challenge is correct, their wearable is a non-expensive technology and their deep-learning algorithms are interesting. The combination of both certainly has the potential (but not more than that at the moment) to revolutionise tracking and diagnosis of neurodevelopmental disorders\delays. Using this rich data can provide new perspectives that even human experts are lacking. Finally, the authors did a very good job in describing and plotting their data and did most of the required statistical tests. It’s definitely shown that the authors worked thoroughly and are very transparent about all their data. This should not be taken for granted. As an advocate of open science, I thank the authors for this and encourage them to do so in the future.

That said, the manuscript needs a lot of work. It’s not clear what this paper is about, no clear questions or hypotheses regarding human development, and the introduction and discussion are a total mess. All these issues make it hard to evaluate the current contribution (besides the methodology, see below for more explanation) of this work to the research of human development, neurodevelopmental disorders, or medicine. Thus, the current form of the paper is unpublishable (and even hard to review), despite its importance.

Major Issues:

Q3.1. The biggest issue for me with this paper is that I do not understand what it is about and therefore, I’m having a hard time evaluating its contribution.

Is it about the novel technology? If so – the authors should publish a method paper in which they provide all the details about the technology and only show a comparison between their tool to the manual annotations. All other discussions, data, figures, etc. are not relevant and should be dropped. The authors do not really show this is effective for the early diagnosis of developmental disorders. It’s not even clear why it should be at the moment, and what developmental disorders? and data from 4 atypical developing children does not prove it. Currently, the authors just bombarded the reader with a lot of details that do not even make sense. For example, why did the authors' selected movements that have so low inter-raters agreement? What can we say about these movements if even two humans cannot agree on what they are? Why are these checked in these fancy algorithms? What is the motivation for all of it? For now, it just seems like the authors tried their (really cool) algorithm on several postures and movements that work but no sense in terms of developmental cascades, neurodevelopmental disorders, etc.

A3.1. The aim of the article is to show development of a holistic framework for the automatic quantification of infant’s motor abilities. It will need many more studies in the future to explore and validate the utility of this methodology in the diverse user cases, such as the actual diagnostics of neurodevelopmental disorders.

Please see our responses to many comparable comments by referees #1 and #2.
We have now removed the data points about atypical infants (Figure 4B; see also our response A1.5.)

A1.5. Thank you for the thoughtful comment. We agree that the presented data regarding longitudinal and abnormal data points is not substantial enough to make real conclusions based on them. Furthermore, their visual emphasis might dissuade readers from understanding the reason for their inclusion (which in our original view was to serve as “sanity checks”), and make false conclusions about the intent of our paper. Based on these reasons, we have omitted the outlier data points in Fig 4B, and removed the visualization of longitudinal connections. As the use of the longitudinal data points in itself is not a problem in this case (see A1.4), they are still included in the Fig 4B.

Regarding the categories of low inter-rater agreement: Please note that there was a consistent reasoning behind the categories; They needed to be intuitive for human understanding and interpretation, as well as being able to pervasively describe all time periods of independent infant activity. In the realistic case with human infants, it implies that the distributions of some postures and/or movement types are dependent on what the infants naturally do. Therefore, that some categories will be rare (e.g. bottom shuffling, or side leaning) in any infant population, but for the sake of completeness they need to be included in the description scheme. Selective addition of such movements into our movement classifier training dataset would be expected to yield a higher performance for the under-performing categories, but this was not within the scope of the present study.

Q3.2. I agree with the authors that it is very powerful to get movement data in a free environment, where infants are not constrained to the lab. Moreover, rich data from daily activities can really contribute to our understanding of infants’ motor development. But the authors do none of these. They do not describe the environment in which these movements occurred, they do not motivate any of the movements\postures they selected (just to clarify - they do motivate but not in terms of free, real-world environment, why are that specific movements\posture will be interesting to test with the wearables in the real-world environment rather than with videos in the lab?). Same for the neurodevelopmental delays – it is true that potentially these wearables can be used to detect delays. But how? The authors did not really test it, and did not associate the movements\postures to specific developmental delays (Again, testing few children with delays is not publishable, they should make a real experimental study).

A3.2. The environment, as described in the “Participants and recordings” section, was either at home or at the research facility within a home-like environment. I.e., the most natural environment (or its approximation) where infants spend their time. The selected postures and movements are motivated by the fact that 1) they are human interpretable 2) are captured by the movement sensors (see Fig 3A) 3) can be used to adequately describe all time periods of infant movement 4) is simple enough, but can be later scaled (i.e., no differentiation between “running” and “walking”, but these can be later split from “standing”-”fluent” if need be). We are not aware of any better approach to meet the above criteria, and yet be simultaneously valid at real-world settings (‘real-world’ being defined as a measurement of everyday life with a minimal amount of set-up or calibration required).

Measuring with the movement sensors (as opposed to video) was a choice out of necessity: It is practically and ethically(!) not possible to carry out video recordings in people’s homes, at least not beyond dedicated research projects. In addition, the infants would move continuously between rooms making it very hard to capture their movements appropriately for any analysis, even for human visual annotations.

Regarding detection of developmental delays, we don’t aim to present definitive proof, however Fig 4D is indirectly suggesting that the novel BIMS score might reflect the actual level of motor developments. Validating this idea will need many further studies and larger datasets, as outlined in our Discussion.

Q3.3. The authors did not convince me that the data they get from the wearables and their algorithms provide new information on human development that we do not know from previous work. The authors only convinced me that their technology can capture some postures that humans can rate by looking at a video. This is not a finding that is worth publication in Communications Medicine. It is an outstanding computer-science paper, but this is not a breakthrough in human development. In my opinion, the main reason for that is the use of supervised learning in this paper. By using supervised learning, the authors train the algorithms to capture things they wanted to look at. This is certainly nice and the accuracies are very impressive, but what is the novelty here? Besides the technology, what is the contribution of this paper?

A3.3. Thank you for the positive assessment of the computational aspects of our work. Regarding novelty, we would like to refer to the assessments of the other two reviewers.

Q3.4. The authors' BIMS score does not provide any insights about developmental delays and the authors are not convincing in explaining the rationale behind it and why it is needed. Similar to the other work in this paper – the authors simply show that their measures correlate with existing measures. This is good and encouraging but what is the big advantage of using BIMS? Why is it better than existing methods?

A3.4. We fully agree that our methods need to be validated in prospective trials against many clinically used ways to assess early neurodevelopment. These are now noted in the revised Discussion.

Q3.5. The authors argue that “our work demonstrates a fully functional solution that is already used in a clinical research environment” – this paper does not provide evidence for that. The authors need a real clinical-trial experiment to back up all their arguments in the discussion section.

A3.5. We have substantially clarified the Discussion regarding the needs for future work in the area.

There are several minor issues and many missing references and discussion or previous empirical evidence. However - the authors must fix the above major issues before those can be given. The paper is not focused, there are too many ideas, too many issues and none of them (Besides the technology itself) is not at a level of publication.

Reviewers' comments:

Reviewer #1 (Remarks to the Author):

The authors have made adequate changes to the manuscript in response to my comments and I am now happy with this version of the manuscript. It opens up important new possibilities for early diagnosis and monitoring of infant movement abilities in natural surroundings and therefore must be expected to have very significant impact.

Reviewer #2 (Remarks to the Author):

The revisions provided have addressed my previous concerns.

Reviewer #5 (Remarks to the Author):

The authors did not sufficiently respond to my comment about inter-rater agreement. Some of their agreement are extremely low – elementary movements (44%), transitions (51%), fluent movements (77%). I don't think anything below 80% agreement between raters is publishable. If there is 44% agreement between raters, the coding is meaningless. It means it's completely subjective. I do not understand why to show this as part of the data.

The authors did not sufficiently respond to my comment about the environment – yes, they mentioned that the environment was “either at home or at the research facility within a home-like environment”. However, they did not describe the environment itself, what was in the environment and why the movement are relevant to this environment. Different environments can elicit different behaviours, and it might affect the findings. Moreover, their explanation for selecting the postures as – “they are captured by movement sensors” – this is what they are testing! They need to explicitly and transparently explain in the paper that they look for specific movements that can be captured by the sensors. They also need to elaborate more about what CANNOT be captured by the sensors and have a transparent discussion about it. The way it is currently written in the paper is misleading. I'm not sure I understand their response about the “choice of necessity to measure movements with sensors and not with video” – to what part of my comment this is related to? The entire paper is about the sensors, that was the whole point. I'm not sure I understand what the authors mean.

I'm still not convinced regarding the novelty of using supervised machine learning to capture the movements. Especially where the postures and movements were selected specifically because sensors can capture them.

In sum, the authors did not address most of the key issues I had with the paper in previous iteration. Their work is publishable and should be published but not in this current form. The authors' argument that they built a “a generalizable, scalable, and effective method to measure infants' spontaneous motor ability across all milestone levels of infant motor development from lying supine to fluent independent walking” – is misleading. They did a proof of concept for a novel approach of recording infant and child behaviour and they should discuss the advantages and limitation of this approach transparently (for example – all the part about clinical assessment is not relevant. They can mention it as future research but they still report the MAIJU in a clinical

assessment. Again, I think this is misleading. This novel and interesting work should be reported in a more transparent, clear, accurate, and modest report.

March 31, 2022

Rebuttal letter #2

Re: "Intelligent wearable allows out-of-the-lab tracking of developing motor abilities in infants"

Reviewer #5 (Remarks to the Author):

Comment 1: The authors did not sufficiently respond to my comment about inter-rater agreement. Some of their agreement are extremely low – elementary movements (44%), transitions (51%), fluent movements (77%). I don't think anything below 80% agreement between raters is publishable. If there is 44% agreement between raters, the coding is meaningless. It means it's completely subjective. I do not understand why to show this as part of the data.

Response 1: It is our humble belief that there is some form of misunderstanding present here. The values presented in the confusion matrices are class-specific recall values (and F-score values in Fig 2B), for which the chance-level performance for a multiclass case is $1/N$, where N is the number of categories (i.e., "completely meaningless" or chance-level recall/F1 values for Movement would be on average at 11%, or $\kappa=0$).

For a thoroughly transparent reporting and to avoid further confusions, we have replaced the F1-scores in Fig 2B and in the manuscript to the respective category-specific kappa scores; we have also added a variety of measures of agreement and proportions of different conditions as a separate matrix in the Supplementary Material (Fig S16). Moreover, parts of the manuscript have been revised in the section "Development of a unified, structured scheme for infant motor ability", where kappa scores and the following phrase have been added:

"Note that for chance-level agreement, $\kappa=0$. Though unideal, an inter-rater agreement around $\kappa=0.6$ is typically described as "moderate" ($\kappa<0.6$) or "substantial" ($\kappa>0.6$) [25]. Such agreement rates are common in annotation tasks with naturally ambiguous categories, such as various EEG tasks [26] or general movements assessment [27]."

Please also note that the purpose of our solution is to assign a category of movement and posture for each time frame of 2 seconds. Therefore, it was not possible to discard epochs or classes due to apparently lower agreements; we have rather decided to present the data transparently, clearly and accurately.

We would also like to point out a large body of literature that shows how common it is to see less-than-ideal agreement levels are in scoring tasks with naturally ambiguous categories. This is a key challenge in e.g., coding behavior of any kind at high temporal accuracy:

Neonatal EEG:

<https://pubmed.ncbi.nlm.nih.gov/30383719/>

Interrater agreement was consistently highest for voltage (binary: substantial, $\kappa = 0.783$; categorical: moderate, $\kappa = 0.562$), seizure presence (fair-substantial; $\kappa = 0.375-0.697$), continuity (moderate; $\kappa = 0.481$), burst voltage (moderate; $\kappa = 0.574$), suppressed background presence (moderate-substantial; $\kappa = 0.493-0.643$), delta activity presence (fair-moderate; $\kappa = 0.369-0.432$), theta activity presence (fair-moderate; $\kappa = 0.347-0.600$), presence of graphoelements (fair; $\kappa = 0.381$), and overall impression (binary: moderate, $\kappa = 0.495$; categorical: fair-moderate, $\kappa = 0.347, 0.465$).

Status epilepticus diagnosis (which is often hoped to be 100% due to clinical implications!)
<https://journals.sagepub.com/doi/epub/10.1177/15500594211050492>

In the NCSE group, the interrater agreement for both 2 categories (episodes with ictal discharges and without ictal discharges) and 3 categories (episodes with possible ictal discharges, with definite ictal discharges, and without ictal discharges) was moderate, with Cohen's kappa = .53% and 95% CI [0.37, 0.69], and Fleiss' kappa = .41% and 95% CI [0.25, 0.57], respectively. The interrater agreement of the individual subjects are shown in Appendix D. In the suspected NCSE group, the interrater agreement was poor (Cohen's kappa = 0; Fleiss' kappa = -.08); hence, we did not further analyze the EEG recordings in this group.

General Movement analysis (visual infant movement analysis that resembles MAIJU studies, however GM analysis is a global aggregate and should be therefore much better than MAIJU)

<https://journals.sagepub.com/doi/epub/10.1177/0883073820981515>

Interrater reliability was greatest during the fidgety age ($\kappa = 0.67$).

EEG background grading

<https://pubmed.ncbi.nlm.nih.gov/35239553/>

The overall interrater agreement was fair ($k = 0.35$). Having Critical Care EEG Monitoring Research Consortium nomenclature certification (40.9%) or EEG board certification (70%) did not improve interrater agreement ($k = 0.26$).

In the case of naturally behaving infants with very frequently changing movements (see our Fig S13), there is an unavoidable confusion between adjacent time frames and/or movement classes. Notably, the original frame-level classification accuracy is less informative *per se*, and it was only used for directly comparing annotations from human experts and classifiers. ***In our approach, the frame-level detections are aggregated over the entire recording sessions where frame-level variability is smoothed out (see Figs 3C, S9-S10), in order to derive descriptive motility statistics for each infant.*** These aggregated metrics show a very high consistency between raters and the algorithm (see Fig. 3).

Comment 2: The authors did not sufficiently respond to my comment about the environment – yes, they mentioned that the environment was “either at home or at the research facility within a home-like environment”. However, they did not describe the environment itself, what was in the environment and why the movement are relevant to this environment. Different environments can elicit different behaviours, and it might affect the findings.

Response 2: We certainly agree with the reviewer that the environment may affect child's behavior, and moreover, that a detailed description of the environment(s) is essential in studies that aim to analyze environmental effects on child's behavior. Our present aim was to develop methodology for quantifying infants' motor behavior rather than studying environment-behavior interaction *per se*. In the revised manuscript, we have added a paragraph describing the recording environments (home and lab) in more detail:

“The recording environment was somewhat variable between infants, which might have affected their behavior on top of the situational variance that is naturally present in spontaneous activity. There may be marked differences between homes in terms of physical layout, furniture or child-relevant objects such as toys. However, a child's own home is still the environment that is best known by the given child, hence it may be considered ecologically relevant for studying natural behavior. Some infants could not be recorded at homes for various reasons (e.g., logistics or parent's preference), and they came to our research lab, BABA center (www.babacenter.fi). BABA rooms are relatively large (4x4 meters) with a large window for natural lighting as well as typical household furniture

including table, chairs, book chest, carpet, and age-appropriate toys. While this environment is not equal to a child's own home, our experience has shown that it is natural enough to encourage children in a seemingly normal exploratory behavior. "

Comment 3: Moreover, their explanation for selecting the postures as – “they are captured by movement sensors” – this is what they are testing! They need to explicitly and transparently explain in the paper that they look for specific movements that can be captured by the sensors. They also need to elaborate more about what CANNOT be captured by the sensors and have a transparent discussion about it. The way it is currently written in the paper is misleading.

Response 3: We agree that the manuscript has to be explicit and transparent. The discussion regarding the constraining factors of the motility description scheme has perhaps been too vague and focused on the point that “the postures and movements need to be detectable by the sensors”. This is indeed one aspect, but there are also two other, equally important aspects: 1) the categories need to have an interpretable meaning from purely visual assessment (i.e., the video) in order to be annotatable and 2) each 2-second frame of infants' independent movement needs to have an appropriate category. We have now revised the manuscript section “Development of a unified, structured scheme for infant motor ability” to highlight this point:

“Leveraging MAIJU's full potential in motor ability assessment calls for motor ability descriptors that strike a balance between 1) having a high temporal accuracy for all moments of independent movement, 2) being captured by movement sensors, while also 3) retaining an interpretable general meaning that is visible from visual assessment. Aiming to optimize for these three constraints, we ... “

Furthermore, as suggested by the reviewer, we have now added a separate section into Discussion to ensure to the readership that the classifier results have to be interpreted through the constraints of the motor ability description scheme:

“Yet, it is important to note that this motor ability description scheme was designed with three constraints: it had to support a comprehensive classification of each 2-second epoch of infant's movement into visually recognizable categories that could, at least in theory, be extractable from movement sensor data, and which would be easily interpretable by humans (e.g., clinicians) in order to build trust on any “overall measure of motility” (such as BIMS) by grounding the measure to real-world observable motor phenomena. Our present motor ability descriptors are therefore not informative for attempts to understand other qualities of an infant's behavior, such as intentionality or fine manual operations characteristic of a child's exploratory behavior. “

Comment 4: I'm not sure I understand their response about the “choice of necessity to measure movements with sensors and not with video” – to what part of my comment this is related to? The entire paper is about the sensors, that was the whole point. I'm not sure I understand what the authors mean.

Response 4: This response was to the reviewer's comment

“why are that specific movements/posture will be interesting to test with the wearables in the real-world environment rather than with videos in the lab?).”

We agree, it is no more relevant for the review of this current paper.

Comment 5: I'm still not convinced regarding the novelty of using supervised machine learning to capture the movements. Especially where the postures and movements were selected specifically because sensors can capture them.

Response 5: As our stated goal is to develop methodology for clinical assessment, its real-world needs have to be taken into account. We have discussed in the manuscript that

“It has recently become popular to directly train deep-learning-based classifier algorithms to turn raw signals into high-level categorical outputs, such as diagnostic 48,49 or clinical outcomes^{35,50,51}. A direct clinical diagnostic prediction from the raw data could have been used in our context as well. However, several issues argue against such a strategy. First, a direct prediction from the sensor data would need datasets that are orders of magnitude larger to accommodate the very high complexity in the raw sensor data. Second, a direct prediction would ignore the intermediate and per se interpretable result, the detailed class-wise motor ability quantitation, thereby greatly limiting transparency and flexibility of the approach.”

I.e., from the clinical point of view, the interpretability of the “supervised” categories is a cornerstone of our approach. If a direct BIMS classifier would output some numerical score based on the raw recordings, any meaningful sanity checks or interpretations of the recording would not be possible by, e.g., the clinician.

We developed our phenomenological description of posture and movement categories as a reasoned compromise between our stated constraints rather than using a strictly data-driven approach. Therefore, the experiment with self-supervising learning was essential to know how clearly our annotation categories are actually represented in the recorded signals. If the critique presented in the statement “Especially where the postures and movements were selected specifically because sensors can capture them.” were fully accurate, our methodology would have had us first perform data-driven clustering based on self-supervised learning, and then have us annotate the obtained clusters.

Comment 6: In sum, the authors did not address most of the key issues I had with the paper in previous iteration. Their work is publishable and should be published but not in this current form. The authors' argument that they built a “a generalizable, scalable, and effective method to measure infants' spontaneous motor ability across all milestone levels of infant motor development from lying supine to fluent independent walking” – is misleading. They did a proof of concept for a novel approach of recording infant and child behaviour and they should discuss the advantages and limitation of this approach transparently (for example – all the part about clinical assessment is not relevant. They can mention it as future research but they still report the MAIJU in a clinical assessment. Again, I think this is misleading. This novel and interesting work should be reported in a more transparent, clear, accurate, and modest report.

Response 6: We fully agree that this work must be transparent, clear, and accurate - especially in claims about clinical or other real-world implications. Indeed, we made a large number of changes in our previous manuscript revision, according to the suggestions by the clinical expert referees (#1 and #2).

We disagree with the reviewer that clinical considerations should not be addressed in the paper, because they have set fundamental constraints that have shaped the development of our method (see Response 5).

In the revised Abstract and Discussion, we have added the following clarifications:

Abstract: “Here we present the proof-of-concept development and validation of an infant wearable system ...”

Discussion: "In summary, the present study shows proof of concept that the...."